# Behavioral Exploration: Learning to Explore via In-Context Adaptation

**Andrew Wagenmaker** [1]  **Zhiyuan Zhou** [1]  **Sergey Levine** [1]

## Abstract

Developing autonomous agents that quickly explore an environment and adapt their behavior online is a canonical challenge in robotics and machine learning. While humans are able to achieve such fast online exploration and adaptation, often acquiring new information and skills in only a handful of interactions, existing algorithmic approaches tend to rely on random exploration and slow, gradient-based behavior updates. How can we endow autonomous agents with such capabilities on par with humans? Taking inspiration from recent progress on both in-context learning and large-scale behavioral cloning, in this work we propose *behavioral exploration*: training agents to internalize what it means to explore and adapt in-context over the space of "expert" behaviors. To achieve this, given access to a dataset of expert demonstrations, we train a long-context generative model to predict expert actions conditioned on a context of past observations and a measure of how "exploratory" the expert's behaviors are relative to this context. This enables the model to not only mimic the behavior of an expert, but also, by feeding its past history of interactions into its context, to select *different* expert behaviors than what have been previously selected, thereby allowing for fast online adaptation and targeted, "expert-like" exploration. We demonstrate the effectiveness of our method in both simulated locomotion and manipulation settings, as well as on real-world robotic manipulation tasks, illustrating its ability to learn adaptive, exploratory behavior.

## 1. Introduction

When tasked with achieving some objective in a novel scenario, humans are often able to make a small number of carefully directed attempts, use the information collected from these attempts to infer the correct behavior, and then successfully solve the objective. In other words, humans exhibit fast online *exploration* and *adaptation*—attempting different behaviors to acquire useful information and adjusting their behavior based on this information to solve the objective. A fundamental goal in the fields of robotics and machine learning is to develop autonomous agents which exhibit such fast online exploration and adaptation. While much work has been devoted to achieving this, existing methods often fall short, typically relying on expensive trial-and-error approaches to exploration that require an infeasibly large number of interactions, and slow, gradient-based behavior updates which are unable to quickly adapt behaviors online.

What would successful exploration and adaptation behavior look like for an autonomous agent? As an example, consider telling a robot "hand me a cup". In this scenario, the robot should attempt to pick up whichever cups are in the scene until it successfully picks up the right one. It should not, however, move its arm randomly, try to pick up a plate, or repeatedly pick up the same cup when there are other cups in the scene it has not yet attempted to pick up. Instead, the correct behavior requires first leveraging knowledge of the scene and what it means to "hand me a cup" to direct actions to the most relevant features, and then quickly adapting based on the attempts that have already been made in order to continue attempting novel, potentially correct, actions. Such a semantic, informed approach to exploration, coupled with fast online adaptation, is much more akin to how humans behave and dramatically more efficient than the "uninformed" novelty-seeking exploration strategies that are more often considered in the existing literature.

In this work we take steps towards developing autonomous agents which exhibit such behavior, focusing in particular on settings where we have access to expert demonstration datasets that provide a prior on what behaviors may be "reasonable" for our task of interest. We are inspired by the recent success of in-context learning to enable fast online adaptation in language domains (Brown et al., 2020), and seek to leverage similar in-context learning capabilities in our setting. We propose training a long-context policy to internalize both which actions an expert would take in a given state, and which expert actions are most *exploratory*. To instill the policy with in-context adaptation capabilities,

[1]Department of Electrical Engineering & Computer Science, University of California, Berkeley. Correspondence to: Andrew Wagenmaker <ajwagen@berkeley.edu>.

*Proceedings of the 42$^{nd}$ International Conference on Machine Learning*, Vancouver, Canada. PMLR 267, 2025. Copyright 2025 by the author(s).

Figure 1: Overview of our proposed approach, behavioral exploration. **Left**: Given offline demonstration data $\mathfrak{D}$, we sample a history (History$_t$), state-action pair $(s_t, a_t)$, and future trajectory proceeding from $s_t$ (Future$_t$), and compute the total *coverage* of the states visited in the future combined with those visited in the past. **Center**: We then train a long-context policy $\pi_{\text{BE}}$ to minimize a standard behavioral cloning loss, but we include in $\pi_{\text{BE}}$'s context the history and total coverage, in addition to the state, enabling $\pi_{\text{BE}}$ to learn to infer "in-context" which actions are likely to increase the future coverage given the past. **Right**: At deployment, $\pi_{\text{BE}}$ can then adapt its behavior by conditioning on the past states visited, and we can regulate how "exploratory" it is by setting the desired coverage conditioning value, allowing for effective exploration over the space of behaviors present in the demonstration data.

in addition to the current state, we condition on a context which includes a history of previous observations, and a measure of how exploratory the expert action in this state is given these observations. This enables the model to not only infer which actions an expert is likely to take, but also which expert actions are exploratory and, by feeding its past observations into its context, allows it to quickly adapt its behavior online to attempt *different* behaviors than what it has already attempted. Critically, our approach is trained to be exploratory *over the space of expert behaviors* and thus naturally restricts its exploration to coherent, reasonable behaviors—behaviors an expert is likely to play, and therefore behaviors likely to solve a given task.

To the best of our knowledge, our approach, which we refer to as *behavioral exploration* (BE), is the first to enable learning of data-driven exploratory behavioral policies that can quickly adapt online as they collect new experience, and achieves this with fully offline training and in-context online adaptation. Furthermore, our approach relies on easy-to-train behavioral cloning methodology (as compared to reinforcement learning approaches, that are often much more challenging to train), and can be efficiently scaled to large-scale datasets. Focusing particularly on robotic and continuous control domains, we demonstrate the effectiveness of our approach on a variety of simulated reinforcement learning (RL) and imitation learning benchmarks, as well as a real-world robotic setting, showing that it yields a significant gain over standard behavioral cloning (BC) and RL approaches.

## 2. Related Work

**Exploration in decision-making.** Exploring an unknown environment is a canonical problem in RL and control, and a significant amount of attention has been devoted to it (Stadie et al., 2015; Osband et al., 2016; Bellemare et al., 2016; Burda et al., 2018; Choi et al., 2018; Ecoffet et al., 2019;

Shyam et al., 2019; Lee et al., 2021; Henaff et al., 2022). The majority of these works, however, aim to explore fully online, and do not make use of prior information or offline data, which is the focus of this work. To our knowledge, only several works seek to leverage offline data to learn exploratory behaviors. Li et al. (2023) and Wilcoxson et al. (2024) study the setting where the offline data does not contain reward labels, and utilize the offline data to warm-start an RL algorithm to explore online. Hu et al. (2023) trains policies to maximize random reward functions offline, then deploys these policies to explore online, coupling them with online RL. Though not explicitly tackling the exploration challenge, a variety of works have considered utilizing offline data to speed up online RL in the setting where both offline and online data contain reward labels (Lee et al., 2022b; Zhang et al., 2023; Uchendu et al., 2023; Zheng et al., 2023; Ball et al., 2023; Nakamoto et al., 2024). Compared to these methods, which all rely on gradient-based online RL training, our approach learns online in-context, allowing for much faster adaptation and exploration.

**Learning for fast online adaptation.** Developing learning algorithms that quickly adapt to new observations broadly falls under the domain of meta-learning. In particular, meta-RL aims to learn a policy on some set of train environments that can quickly adapt to a new test environment; see Beck et al. (2023) for a complete overview. A key challenge here is learning to explore effectively in the test environment, and a variety of methods have been proposed to achieve this (Duan et al., 2016; Wang et al., 2016; Mishra et al., 2017; Zintgraf et al., 2019; Rakelly et al., 2019; Kamienny et al., 2020; Zhang et al., 2020; Liu et al., 2021; Norman & Clune, 2023; Jiang et al., 2024). These works require online access to train environments, however, while in contrast we only assume access to offline data. Several works have considered meta-RL from only offline data (Li et al., 2020; Mitchell et al., 2021; Dorfman et al., 2021; Rafailov et al., 2021; Ghosh et al., 2022; Pong et al., 2022), including

a recent line of work that aims to train transformer-style models which learn online adaptation strategies in-context (Laskin et al., 2022; Liu & Abbeel, 2023; Lee et al., 2024), or utilizes the in-context learning abilities of LLMs to solve simple decision-making problems (e.g., multi-armed bandits) (Krishnamurthy et al., 2024; Nie et al., 2024; Monea et al., 2024). While our work is related to meta-RL in that we also aim to learn fast adaptation behaviors, our setting is fundamentally different: all of the above works assume access to a reward function, and seek to learn a policy maximizing this reward function, while we instead assume access to a set of demonstrations, and aim to learn exploratory behaviors. We highlight as well the meta-imitation learning setting, where the goal is to learn a policy that can imitate a very small number of expert demonstrations (Duan et al., 2017; Finn et al., 2017; James et al., 2018; Dasari & Gupta, 2021; Raparthy et al., 2023; Fu et al., 2024). The objective of these works is fundamentally different than ours, however: they aim to *imitate* the "history", while our goal is to behave *differently* than the history.

**Learning control policies from expert demonstrations.** Learning control policies by mimicking expert demonstrations, behavioral cloning, has a long history (Argall et al., 2009; Ross et al., 2011; Bojarski, 2016; Zhang et al., 2018; Rahmatizadeh et al., 2018; Mandlekar et al., 2021). Over the last several years, significant attention has been given to applying BC to real-world control and robotic domains, and much progress has been made in deploying modern learning paradigms (Shafiullah et al., 2022; Cui et al., 2022b; Zhao et al., 2023; Chi et al., 2023; Dasari et al., 2024) and training controllable policies by conditioning on human language and other modalities (Stepputtis et al., 2020; Brohan et al., 2022; Nair et al., 2022; Team et al., 2024; Kim et al., 2024; Black et al., 2024; Gu et al., 2023; Cui et al., 2022a;b; Black et al., 2023; Sundaresan et al., 2024). Another line of work aims to learn "skills" from expert data, and compose these with a high-level skill policy (Ajay et al., 2020; Singh et al., 2020; Pertsch et al., 2021). Our work is somewhat related to this line of work, but instead of training policies to directly mimic an expert in solving tasks as these works do, we focus on training policies that can effectively explore and adapt. We also mention the work of Zhou et al. (2024), which trains a goal-conditioned BC policy and deploys this online to collect data and perform policy improvement. While this work can be seen as tackling the exploration problem, the proposed approach is quite different, relying on a multi-stage system composing several models trained on internet-scale data; in contrast, we aim to learn a single model from only demonstration data that internalizes the ability to explore.

**Decision-making with return-conditioned policies.** Our proposed methodology relies on training a generative model that conditions on a measure of the coverage an action should lead to. This measure of coverage can be thought of

as a "return", and our approach therefore bears resemblance to work on return-conditioned policies for decision-making. Typically, these works condition on the desired reward-to-go (Schmidhuber, 2019; Srivastava et al., 2019; Kumar et al., 2019; Emmons et al., 2021), but have also considered goal conditioning (Ghosh et al., 2019; Ding et al., 2019; Lynch et al., 2020). More modern variants have proposed training higher-capacity transformer or diffusion models (Chen et al., 2021; Furuta et al., 2021; Ajay et al., 2022; Lee et al., 2022a; Huang et al., 2024; Wu et al., 2024; Schmied et al., 2024; Yan et al., 2024). While our work bears resemblance to these works methodologically, our focus is quite different—rather than training a model to maximize reward, we train the model to explore, requiring a very different return structure and much more intricate history dependence.

## 3. Preliminaries

We consider decision-making in the context of reward-free Markov decision processes (MDPs), which we denote by a tuple $\mathcal{M} := (\mathcal{S}, \mathcal{A}, P, p_0)$, where $\mathcal{S}$ is the set of states, $\mathcal{A}$ the set of actions, $P : \mathcal{S} \times \mathcal{A} \rightarrow \triangle_{\mathcal{S}}$ the transition kernel, and $p_0 \in \triangle_{\mathcal{S}}$ the initial state distribution. A trajectory denotes an interaction sequence where we sample an initial state $s_0 \sim p_0$, take some action $a_0$, transition to state $s_1 \sim P(s_0, a_0)$, and this process continues; for simplicity, we assume each trajectory is of length $K$. Throughout we use $\boldsymbol{\tau}$ to denote such a trajectory, $\boldsymbol{\tau} = (s_0, a_0, s_1, a_1, \ldots, s_K, a_K)$, and we let $\boldsymbol{h} \subseteq S^H$ denote some set of states—typically this will correspond to the set of previous trajectories an agent has observed, its history. We denote a policy by $\pi : \mathcal{S} \times \mathcal{X} \rightarrow \triangle_{\mathcal{A}}$, where $\mathcal{X}$ could be some other conditioning space (in particular, we will condition $\pi$ on its history of past interactions, $\boldsymbol{h}$, as we describe below). We say that a trajectory $\boldsymbol{\tau}$ is generated by policy $\pi$ if $a_k \sim \pi(s_k, x_k)$ for each step $k$ in $\boldsymbol{\tau}$.

**Demonstration data.** We assume access to a demonstration dataset $\mathfrak{D} = \{\boldsymbol{\tau}^t\}_{t=1}^T$, where for each $t$, $\boldsymbol{\tau}^t := (s_0^t, a_0^t, \ldots, s_K^t, a_K^t)$ denotes a trajectory collected by some behavior policy $\pi_\beta$ in $\mathcal{M}$ (note that $\pi_\beta$ may be a mixture policy over multiple demonstrators). We do not make explicit assumptions on the optimality of the behavior of $\pi_\beta$, but we do assume that the behaviors exhibited by $\pi_\beta$ are a reasonable prior over potentially "useful" behaviors, as we formalize in the following.

## 4. Learning to Explore from Demonstrations

Given demonstration data $\mathfrak{D}$, our goal is to learn a policy that, when deployed in a test environment, explores its setting and adapts its behavior online in order to continue to explore. In particular, rather than exploring uniformly over the space of all possible behaviors, we would like the ex-

ploration to be focused over the space of "useful" behaviors represented in $\mathfrak{D}$. In this section we formalize this objective, and outline our proposed approach.

### 4.1. Exploration Over Demonstration Coverage

Canonically, the goal of exploration is to interact with an environment in order to collect data that achieves high *coverage*—visiting as many useful and informative states as possible. To formalize this, assume that we have access to some featurization of our environment, $\phi(s) \in \mathbb{R}^d$. For example, in a finite-state setting we might have $\phi(s) = e_s$, in a continuous-state setting $\phi(s)$ might correspond to a vectorized representation of the state (e.g., the final layer of a neural network state encoder), or in an image-based setting $\phi(s)$ could correspond to a feature vector of our image given by some image feature extractor. In short, $\phi(s)$ denotes some mapping which allows us to relate states based on their vector representations, and we can therefore quantify coverage in terms of the degree to which this representation space is spanned. For the purposes of this work, we take inspiration from classical estimation theory, and propose quantifying coverage via the trace of the inverse feature matrix[1]. Precisely, given some set of states $h := \{s_i\}_{i=1}^t$, letting $\Lambda(h) := \sum_{i=1}^t \phi(s_i)\phi(s_i)^\top$, we define the coverage of $h$ as:

$$\mathsf{cov}(h) := \frac{1}{\mathrm{tr}\big((\Lambda(h) + \lambda \cdot I)^{-1}\big)}. \tag{1}$$

In the finite-state setting, for example, this reduces to $\mathsf{cov}(h) = (\sum_s \frac{1}{N(s;h)+\lambda})^{-1}$ for $N(s;h)$ the number of times state $s$ appears in $h$, recovering a standard notion of total state coverage. In more general, potentially infinite-state, settings, (1) corresponds to how well we span the entire feature space.

Standard exploration approaches would typically aim to explore uniformly over the space, collecting data with the goal of (approximately) maximizing (1). While achieving high uniform coverage would provide a rich dataset for future downstream learning, for many problems of interest the entire reachable space is extremely large, and achieving such coverage may be infeasible or, at the very least, extremely wasteful. Instead, we would like to focus our exploration on only the most relevant parts of the space. If we have access to demonstrations from some policy $\pi_\beta$ which exhibits behaviors spanning the space of "useful" behaviors, this gives

us a prior over where to focus exploration, and, rather than exploring uniformly, we might focus our exploration on the space of behaviors spanned by $\pi_\beta$. For instance, returning to the example given in the introduction, instead of exploring all possible robot state configurations—the behavior that would result when maximizing (1)—we should explore only within the space of motions represented in a dataset of expert robot demonstrations, attempting different manipulation behaviors likely to solve our task. More generally, by focusing our exploration on the behaviors present in an effective demonstration policy $\pi_\beta$, we can efficiently collect highly targeted data that could be utilized for downstream learning objectives, or, if $\pi_\beta$ exhibits diverse task-solving behaviors, explore over these behaviors to arrive at a solution to a task of interest.

To formalize coverage over behaviors in $\pi_\beta$, denote $\Lambda_\beta := \mathbb{E}^{\pi_\beta}[\sum_{k=1}^K \phi(s_k)\phi(s_k)^\top]$, $U_\beta \Sigma_\beta U_\beta^\top$ the singular-value decomposition of $\Lambda_\beta$, and $U_{\beta,\epsilon} \in \mathbb{R}^{d \times r}$ the matrix of singular vectors of $\Lambda_\beta$ corresponding to singular values of at least $\epsilon$. $U_{\beta,\epsilon}$ then corresponds to the span of features covered by $\pi_\beta$ (with weight at least $\epsilon$) and, augmenting our uniform coverage metric, (1), to:

$$\mathsf{cov}_\beta(h) := \frac{1}{\mathrm{tr}\big((U_{\beta,\epsilon}^\top \Lambda(h) U_{\beta,\epsilon} + \lambda \cdot I)^{-1}\big)}, \tag{2}$$

we can quantify how well $h$ covers the space of features spanned by $\pi_\beta$ with $\mathsf{cov}_\beta(h)$. For example, in the finite-state setting, $\mathsf{cov}_\beta(\cdot)$ corresponds to the coverage only over states visited by $\pi_\beta$ with probability at least $\epsilon$, rather than coverage over all states, or in the robotic manipulation setting, this may correspond to coverage over all coherent manipulation motions. Critically, achieving coverage over $U_{\beta,\epsilon}$ could be much easier than achieving coverage over the entire feature space, and allows us to focus our exploration only on the most salient aspects of the environment. Given this, we propose the following objective, which formalizes our goal of exploring over the behaviors exhibited by $\pi_\beta$.

**Objective 4.1** (Behavior Policy Coverage). *For $\mathsf{cov}_\beta(\cdot)$ the coverage over the features spanned by the behavior policy $\pi_\beta$, (2), and $\Lambda_\pi := \mathbb{E}^\pi[\sum_{k=1}^K \phi(s_k)\phi(s_k)^\top]$, use $\mathfrak{D}$ to find a policy $\pi$ which maximizes $\mathsf{cov}_\beta(\Lambda_\pi)$.*

### 4.2. Learning Exploratory Behaviors via Conditional Distribution Modeling

A natural choice for a policy maximizing coverage over the space spanned by $\pi_\beta$ would be $\pi_\beta$ itself—by definition, $\pi_\beta$ covers the space spanned by $\pi_\beta$—

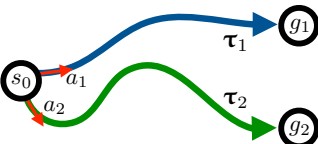

Figure 2: Illustrative example.

which suggests simply training a BC policy on $\mathfrak{D}$ to solve

---

[1]For example, the trace of the inverse information matrix, which we can think of $\sum_{i=1}^t \phi(s_i)\phi(s_i)^\top$ as approximating, corresponds to the estimation error of the maximum likelihood estimator (Van der Vaart, 2000), and, by the Cramer-Rao lower bound, lower bounds the estimation error of any unbiased estimator (Pronzato & Pázman, 2013). Furthermore, this expression is standard in the experiment design literature; maximizing (1) is commonly known as *A-optimal experiment design* (Pukelsheim, 2006).

Objective 4.1. To illustrate the shortcomings of BC, however, consider Figure 2. In this example, assume that we start at state $s_0$ and that there are two goals of interest, $g_1$ and $g_2$, both within the coverage of $\pi_\beta$. At $s_0$, we can either take action $a_1$, which will lead us along trajectory $\boldsymbol{\tau}_1$ to $g_1$, or $a_2$ which will lead us along $\boldsymbol{\tau}_2$ to $g_2$. Assume that $\pi_\beta(a_1 \mid s_0) = 1 - p$ and $\pi_\beta(a_2 \mid s_0) = p$, for $p \ll 1$. If we clone $\pi_\beta$, since $\pi_\beta$ only takes $a_2$ with small probability, the vast majority of the time we will take $a_1$ and follow $\boldsymbol{\tau}_1$ to $g_1$, foregoing the opportunity to reach $g_2$. In contrast, if we were to take $a_1$ and $a_2$ with equal probability, we would much more quickly reach both goals. Thus, while $\pi_\beta$ will *eventually* achieve coverage over $g_1$ and $g_2$, we can achieve this much more efficiently with a different weighting of actions.

While naïve BC is therefore suboptimal, we see that it *does* place mass on both actions of interest and, if we could reweight this distribution in some way in order to increase mass on $a_2$, it may offer a feasible method to collect high-coverage data. Let us assume that we are in $s_0$ and, on the previous trajectory, already took $a_1$ and observed $\boldsymbol{\tau}_1$—denote by $\boldsymbol{h}$ our history of past observations, $\boldsymbol{h} = \boldsymbol{\tau}_1$. In this case we would expect $\mathsf{cov}(\boldsymbol{h} \cup \boldsymbol{\tau}_2) \gg \mathsf{cov}(\boldsymbol{h} \cup \boldsymbol{\tau}_1)$—since $\boldsymbol{\tau}_2$ covers states not previously encountered in $\boldsymbol{h}$, it would, in general, increase coverage relative to $\boldsymbol{h}$ more than observing $\boldsymbol{\tau}_1$ again. If we let $P^{\pi_\beta}[\cdot]$ denote the distribution over actions induced by $\pi_\beta$[2] and $\boldsymbol{\tau}$ the trajectory that results from taking actions under $\pi_\beta$ from state $s_0$, then we see that

$$P^{\pi_\beta}[a_2 \mid s_0, \boldsymbol{h}, \mathsf{cov}(\boldsymbol{h} \cup \boldsymbol{\tau}) = \mathsf{cov}(\boldsymbol{h} \cup \boldsymbol{\tau}_2)]$$
$$\gg P^{\pi_\beta}[a_1 \mid s_0, \boldsymbol{h}, \mathsf{cov}(\boldsymbol{h} \cup \boldsymbol{\tau}) = \mathsf{cov}(\boldsymbol{h} \cup \boldsymbol{\tau}_2)].$$

In other words, by conditioning on the subsequent trajectory achieving high coverage relative to $\boldsymbol{h}$, the likelihood of $\pi_\beta$ taking $a_2$ and therefore reaching $g_2$ increases significantly, as $a_2$ is much more likely to increase coverage than $a_1$. The following result makes this formal, and shows that (in certain cases), this history-dependent conditional distribution over actions is in fact an optimal solution to Objective 4.1.

**Proposition 4.2** (Informal). *Assume that we are in a deterministic environment, that for any terminal state $s_K$, $\boldsymbol{\phi}(s_K) \in \{\boldsymbol{e}_1, \ldots, \boldsymbol{e}_{d-1}\}$, and that $\boldsymbol{\phi}(s) = \boldsymbol{e}_d$ for $s$ not a terminal state. Let $\boldsymbol{h}_t$ denote the history of states visited up to trajectory $t$. Then under several additional technical assumptions, the policy which takes actions at trajectory $t$ with probability*

$$P^{\pi_\beta}[\,\cdot\mid s, \boldsymbol{h}_t, \mathsf{cov}(\boldsymbol{h}_t \cup \boldsymbol{\tau}) = \max_{\boldsymbol{\tau}'} \mathsf{cov}(\boldsymbol{h}_t \cup \boldsymbol{\tau}')] \quad (3)$$

*is an optimal solution to Objective 4.1.*

Thus, while naïve behavioral cloning is suboptimal, Proposition 4.2 shows that by setting our policy to a carefully

---

[2]More precisely, if we consider the distribution over trajectories induced by rolling out $\pi_\beta$ on $\mathcal{M}$, then $P^{\pi_\beta}$ denotes the marginal of this distribution over actions.

conditioned distribution over the actions of the behavior policy, we can arrive at an optimal solution to Objective 4.1. Critical to achieving this is that we condition on the history and *adapt* our behavior based on the history in order to increase coverage—as the previous example illustrates, only by adapting to our history can we ensure the exploratory action is taken. We note that this result only requires conditioning on cov and not on $\mathsf{cov}_\beta$, thereby avoiding explicit dependence on $\boldsymbol{U}_{\beta,\epsilon}$—the policy of (3) explores over $\boldsymbol{U}_{\beta,\epsilon}$ by construction since its action distribution is just a reweighting of $\pi_\beta$. Please see Section A for the proof of Proposition 4.2.

To learn a policy from $\mathfrak{D}$ which explores over the space of features spanned by $\pi_\beta$, the above argument motivates the following simple supervised learning problem, which we refer to as *behavioral exploration*.

**Behavioral Exploration** (BE). *Given expert demonstration data $\mathfrak{D}$, find a policy $\pi_{\mathsf{BE}}$ maximizing the following:*

$$\sum_{t=1}^{T} \sum_{k=1}^{K} \mathbb{E}_{\boldsymbol{h} \sim \mathcal{H}(\mathfrak{D})}[\log \pi_{\mathsf{BE}}(a_k^t \mid s_k^t, \boldsymbol{h}, \mathsf{cov}(\boldsymbol{h} \cup \boldsymbol{\tau}_k^t))], \quad (4)$$

*where $\boldsymbol{\tau}_k^t := (s_k^t, \ldots, s_{K_t}^t)$ denotes the subtrajectory of $\boldsymbol{\tau}^t$ starting at step $k$, and $\mathcal{H}(\mathfrak{D})$ some distribution over demonstration trajectories.*

We can think of this objective as corresponding to the maximum likelihood estimate of the conditional distribution given in (3) (see e.g. Foster et al. (2024)). We emphasize that $\pi_{\mathsf{BE}}$ naturally focuses its exploration on the space spanned by $\pi_\beta$ since, ultimately, it is still fitting the action distribution induced by $\pi_\beta$.

Given a policy $\pi_{\mathsf{BE}}$ solving (4), at deployment, if we are at step $k$ and state $s_k$, and have already visited states $\boldsymbol{h}_k$, we sample our next action as $a \sim \pi_{\mathsf{BE}}(\cdot \mid s_k, \boldsymbol{h}_k, \mathsf{exp})$, where $\mathsf{exp} \in \mathbb{R}$ is a measure of how exploratory the sampled action should be relative to $\boldsymbol{h}_k$. If we set $\mathsf{exp}$ to be large, then this will increase the likelihood of sampling actions that lead to higher coverage data, while if $\mathsf{exp}$ is small, the sampled action is incentivized to induce behaviors close to the already collected observations.

### 4.3. Practical Implementation

The behavioral exploration objective, (4), forms the basis for our proposed approach. Here we make several comments on the practical implementation of BE.

**Architectural considerations.** Our goal in (4) is to learn a policy modeling a potentially complex and multi-modal conditional distribution, in particular in control settings with continuous actions. Furthermore, we must condition on a significant amount of information—the current state as well as a history of previous states. Effectively modeling this distribution therefore requires an architecture able to handle

such complex continuous distributions, as well as a large conditioning space. We propose utilizing a diffusion model with a transformer backbone to address these challenges. Diffusion models are known to effectively handle complex, continuous, multi-modal distributions in policy learning settings (Chi et al., 2023), and the long-context capabilities of transformers allow us to efficiently capture the large conditioning space. In practice, for all experiments we use the state-of-the-art transformer-based diffusion policy proposed in Dasari et al. (2024). We allocate a single token to each conditioning variable—the current state, coverage-to-go, and each state in the history—enabling the model to infer the relationship between these modalities, and at deployment adapt its behavior in-context based on the observed history.

**Selecting the history distribution.** At deployment, our history $h$ may contain a wide distribution of different states, corresponding to those encountered by the policy online, and we would like the policy to infer, for any such set of states, where it should direct its exploration. As such, we should ideally choose the training history distribution, $\mathcal{H}(\mathfrak{D})$, to correspond to the induced distribution of online states. Estimating this induced distribution is challenging, however, and requires solving a fixed-point problem. We note, though, that this distribution is simply a reweighting of the distribution induced by $\pi_\beta$. Given this, in practice we simply choose $\mathcal{H}(\mathfrak{D})$ to be a uniform distribution over trajectories in $\mathfrak{D}$, thereby enabling $\pi_{\text{BE}}$, in the ideal case, to learn exploratory behaviors over all histories it is likely to encounter.

**Incorporating task conditioning.** Assume that for each $\boldsymbol{\tau}^t$ we have a corresponding task label, $y^t$—for example, a language command. We can easily modify (4) to incorporate such task labels by simply conditioning on $y^t$ as well:

$$\sum_{t=1}^{T} \sum_{k=1}^{K} \mathbb{E}_{\boldsymbol{h} \sim \mathcal{H}(\mathfrak{D})}[\log \pi_{\text{BE}}(a_k^t \mid s_k^t, y^t, \boldsymbol{h}, \text{cov}(\boldsymbol{h} \cup \boldsymbol{\tau}_k^t))].$$

Maximizing this objective enables us to restrict our behavior to within the space of a given task, thereby further focusing our exploration to the most relevant parts of the space.

# 5. Experiments

In our experimental evaluation, our focus is on understanding (a) whether BE is able to learn effective exploration strategies from offline demonstration data and adapt quickly online, (b) if BE is able to effectively focus its exploration over the space of behaviors present in the demonstration data, and (c) if BE scales to large-scale, real-world imitation learning (IL) settings. Since our method combines elements of both RL and IL, we aim to show that it can perform well in both settings. We first focus on RL benchmarks, where we compare against RL-based approaches to exploration, and then on IL, where we consider both simulated and real-

world robotic tasks. Additional details on all experimental settings are given in Section B.

## 5.1. Fast Exploration in RL Benchmarks

For our RL experiments, we evaluate BE on a subset of the environments in the D4RL benchmark (Fu et al., 2020), focusing in particular on settings that require exploration. For each D4RL setting, a standard dataset exists to enable offline learning; we utilize these datasets to train our approach, and evaluate online. Notably, the data in these datasets is from a scripted policy and is not optimal. While D4RL is primarily an offline RL benchmark, a variety of recent works have used it to benchmark online learning and exploration approaches initialized from offline data (Zhang et al., 2023; Uchendu et al., 2023; Li et al., 2023; Wilcoxson et al., 2024). Our focus in this section will be to compare with such RL approaches. In particular, we consider the setting proposed by Li et al. (2023) and Wilcoxson et al. (2024), where reward labels are removed from the offline data and the agent must explore online in order to find reward; we note that standard offline RL and offline-to-online RL algorithms do not directly apply here as they require reward information offline.

**Environment details.** We focus our experiments on the state-based Antmaze and Kitchen environments from D4RL, where the tasks, respectively, are to navigate an ant agent in a maze in order to find a goal location, and use a robotic arm to accomplish various tasks in a kitchen. For Antmaze, we evaluate on the `medium` and `large` variants of the maze using the `diverse` offline dataset, and for each test with four distinct goal locations—which are initially unknown to the agent and must be found by exploring the maze—utilizing the same goal locations as are used by Wilcoxson et al. (2024). For Kitchen, we utilize the `partial` variant of the offline data. As our objective is to evaluate the ability of our approach to explore and adapt online, rather than the standard RL objective of learning an optimal policy, we modify the standard setup somewhat. First, for both environments, as a metric of success we consider the time the agent takes to achieve each task at least once, and for Antmaze we also evaluate maze traversal ability, measuring this in terms of the number of regions reached (where we define a region to be a block in the maze grid, see Figure 12). Second, to increase the challenge of exploration and test fast-adaptation ability, we shorten the episode length in the Antmaze environment and evaluate on a relatively short number of total environment steps compared to standard RL evaluations (20k and 3k for Antmaze and Kitchen, respectively).

**Benchmark algorithms.** We compare against four representative algorithms from those evaluated in Wilcoxson et al. (2024), as well as a BC baseline.

Online RND: The RND algorithm (Burda et al., 2018) is a canonical approach for exploration that incentivizes ex-

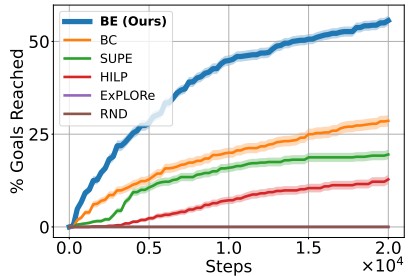
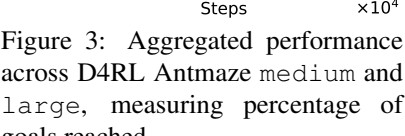
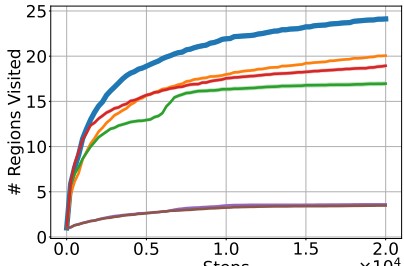
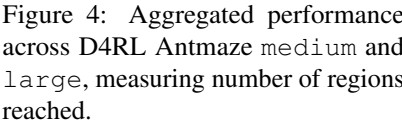
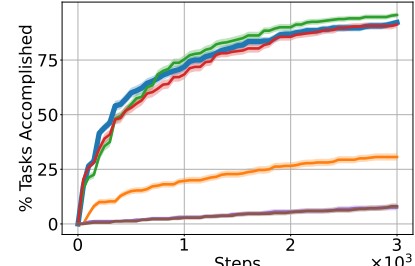

Figure 3: Aggregated performance across D4RL Antmaze `medium` and `large`, measuring percentage of goals reached.

Figure 4: Aggregated performance across D4RL Antmaze `medium` and `large`, measuring number of regions reached.

Figure 5: Performance on D4RL Kitchen, measuring percentage of tasks accomplished.

ploratory behavior by labeling observations with a pseudo-reward intended to reflect uncertainty. To achieve this, a neural network is randomly initialized and, online, the agent trains another network to match the predictions of this network over the observed states. The pseudo-reward is the mismatch between the predictions of the networks; intuitively, unvisited states will have larger prediction error, and therefore pseudo-reward, than visited states. We run RND using standard online RL, ignoring the offline data completely.

ExPLORe: The ExPLORe algorithm (Li et al., 2023) is a variant of online RND, but makes use of the offline data as well. This approach initializes an online RL algorithm by feeding the offline data into its replay buffer, and labels this offline data with the RND rewards.

HILP: HILP is an offline unsupervised skill discovery method recently proposed by Park et al. (2024). It first trains a set of "skill" policies on the offline data, learning skills that efficiently traverse a learned representation space. Skills are indexed by a latent vector $z$, which can be utilized as a transformation of the action space. That is, rather than taking actions in the original action space, we can think of $z$ as the action, and execute it by playing the corresponding skill policy. We take this approach for the HILP baseline, running an online RL algorithm over the skill space. In particular, we use the "HILP with offline data" variant given in Wilcoxson et al. (2024), which feeds the offline data into the replay buffer of the online RL algorithm.

SUPE: SUPE, proposed by Wilcoxson et al. (2024), is another skill-based approach. Similar to HILP it trains a skill policy on the offline data, and then online learns a policy over skill space. In addition, SUPE utilizes RND rewards to incentivize exploration online, and feeds the offline data, labeled with the RND rewards, into its replay buffer.

BC: As a final baseline, we compare against standard BC. We utilize the same diffusion policy base as is utilized by our method, conditioning only on state. At deployment, we simply run this learned policy online.

For all methods which require offline training, we utilize

the checkpoints provided by Wilcoxson et al. (2024) (which are trained on the provided offline datasets with reward labels removed), and run each method exactly as described in Wilcoxson et al. (2024). We train and run BE as described in Section 4.2, training with 8 different random seeds on the provided offline datasets, and in online deployment feeding the data collected in previous episodes into the learned policy's context. We remark that BE relies only on in-context online adaptation, while all other baselines considered here (with the exception of BC) rely on more expensive online gradient-based updates. All methods are evaluated online with 80 trials (10 per offline seed). Error bars denote 1 standard error.

**Results.** Our results on D4RL are illustrated in Figures 3 to 5. On Antmaze, BE reaches the goals much more quickly and achieves significantly higher coverage than the considered baselines. In particular, we note that the entire possible search space of Antmaze is significantly larger than simply covering the $(x, y)$-locations in the grid—we can also attempt to cover the entire space of possible ant configuration states. However, the demonstration data contains a much lower-dimensional space of ant configurations, corresponding to standard walking motions, and we observe that BE focuses its exploration on such behaviors, ultimately enabling effective exploration over the $(x, y)$-space.

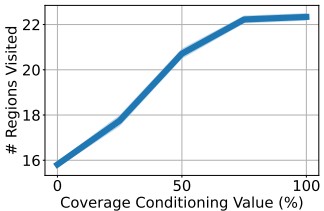

Figure 6: Calibration of BE policy on Antmaze `medium`.

In Figure 6 we further illustrate the *calibration* of the learned BE policy, plotting the total number of regions visited against the coverage value we condition on at deployment—as this illustrates, the policy has learned calibrated behavior and is able to adjust the degree to which it explores by tuning the coverage value. On Kitchen, our approach performs comparably to the state-of-the-art skill-based RL approaches. We emphasize that we are able to achieve this via a simple supervised-learning based-

approach, coupled with a policy that adapts online in-context, in contrast to the more complex RL training of SUPE and HILP. Furthermore, our approach yields significant improvements over BC, the standard non-adaptive supervised learning approach.

## 5.2. Exploration in Imitation Learning

We next consider simulated and real-world vision-based imitation learning settings. In simulation, we utilize the Libero benchmark (Liu et al., 2024), which simulates a variety of robotic manipulation and pick-and-place tasks, while in the real world, we train a policy for object manipulation on the Bridge dataset (Walke et al., 2023), and evaluate on the WidowX robot. Both settings cover a wide range of environments and tasks, and demonstrate the ability of our approach to learn from such diverse data. Our focus here is to test the ability of our method to explore over the space of expert behaviors represented in the training data, while still learning useful task-solving behaviors.

### 5.2.1. SIMULATED RESULTS ON LIBERO BENCHMARK

**Environment details.** The Libero benchmark contains a variety of scenes, simulating different robotic manipulation and pick-and-place tasks. Figure 10 visualizes 4 such scenes—example tasks in the lower left scene include "turn on the stove and put the frying pan on it" or "put the moka pot on the stove" (see Section B.2

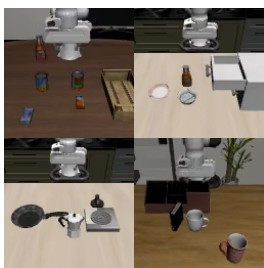

Figure 10: Example Libero tasks.

for additional task descriptions). We run all experiments on the Libero 90 dataset, which includes 90 tasks spread across 21 distinct scenes. For each task, the dataset provides 50 human demonstrations of successful completion, which we utilize as our demonstration dataset. We consider two evaluation settings. In the first, we choose a task we would like the agent to solve, but do not instruct the policy which task we have chosen. As each scene contains from 2-7 different tasks, to successfully solve the hidden task, the agent must attempt and successfully complete different tasks until it has completed the hidden one. This simulates settings where the goal is ambiguous and we would like the agent to explore possible solutions until it arrives at the correct one. We count a trial as a success if the hidden task has been solved at least once across 15 attempts. In our second evaluation, we do provide a task designation, and evaluate the ability of each policy to solve the task after 6 attempts, testing the ability of a policy to try diverse behaviors for a given task.

**Benchmark algorithms.** As Libero is primarily an imita-tion learning benchmark, we focus on comparing with other imitation learning approaches. We compare against a BC baseline, which utilizes the same diffusion policy base as our approach, and several variants of standard BC: BC with action noise, where we add a small amount of Gaussian noise to the actions produced by the BC policy, and BC with history conditioning, where we train a BC policy conditioned on a history of past observations. For the hidden-task evaluation, we also consider deploying a task-conditioned BC policy with a randomly sampled task at each attempt (simulating the use of task-conditioned policy in settings where the task is ambiguous) and SUPE, the best-performance RL baseline considered in the previous section. We again deploy BE as described in Section 4.2, feeding observations from the past episodes collected online into its context. For each method considered here, with the exception of SUPE, we train 5 policies with different random seeds. For each policy, we evaluate on each of the 90 tasks in the benchmark 3 times. For SUPE we evaluate it once on each of the 90 tasks.

**Results.** Our results are given in Figures 7 and 8. BE much more quickly solves the correct task in the hidden-task evaluation—achieving a 2-3× reduction in the number of attempts required compared to standard BC—and similarly solves the task more quickly (and achieves a higher final success rate) than BC in the evaluation with task conditioning. This demonstrates the ability of BE to explore over the space of behaviors represented in the expert demonstrations—instead of exploring randomly, BE explores by selecting *different* task-solving behaviors from the demonstration dataset. This exploration does not hurt its ability to successfully solve tasks, but instead enables BE to attempt diverse behaviors likely to solve the task, achieving a higher task success rate than BC. We see that this is in contrast to augmenting BC with action noise. While BC with action noise does induce some amount of additional exploration, improving the BC policy's success rate in the hidden-task evaluation, action noise significantly hurts BC's performance in the evaluation with task conditioning, likely due to the action noise causing the behavior to deviate too far from the demonstrated task-solving behavior. This highlights the critical advantages of behavioral exploration over random exploration in inducing behaviors that are exploratory, but which still lead to high task success rates. Similarly, we see that naïvely conditioning on history can significantly hurt BC performance, a phenomenon that has been observed in the existing literature (De Haan et al., 2019), yet the history conditioning employed by BE enables $\pi_{\mathrm{BE}}$ to utilize the history to actually *improve* its performance.

### 5.2.2. REAL-WORLD RESULTS ON WIDOWX ROBOT

For our real-world experiments, we train a policy on the BridgeData V2 dataset (Walke et al., 2023), which contains a diverse set of over 60,000 robotic teleoperation demon-

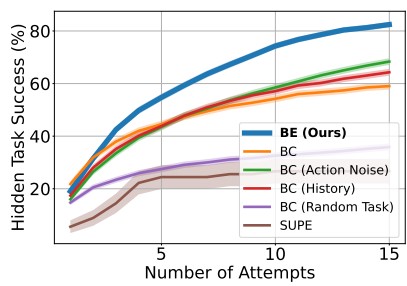

Figure 7: Performance on Libero in evaluation with task hidden.

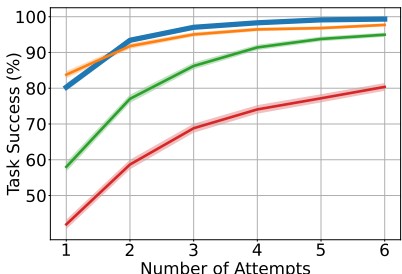

Figure 8: Performance on Libero in evaluation with task provided.

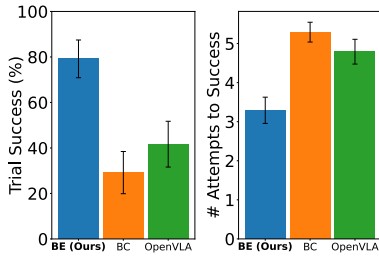

Figure 9: Performance on WidowX robot in reaching evaluation.

strations across a wide range of environments and tasks. This setting demonstrates the ability of our approach to scale to real-world deployment and large-scale vision-based datasets, and learn from such diverse demonstration data.

**Environment details.** We evaluate our policy on the WidowX 250 6-DoF robot arm. We consider 3 different reaching tasks, where for each task, we place 2 objects in front of the robot, and the goal is to interact with each object in some way (see Figure 11). A "trial" consists of 5 consecutive episodes in the same scene—we count a trial as a success if the policy interacts with both objects at least once across the 5 attempts

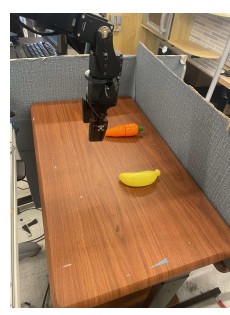

Figure 11: Example WidowX setup.

(and consider any instance of the robot's end-effector making contact with one of the objects as a successful "interaction"). Similar to Libero, this simulates settings where the robot is given an ambiguous task specification, and we would like it to attempt different behaviors to arrive at the correct one.

**Benchmark algorithms.** We again compare against a BC baseline, using the same base diffusion model as our method. We also compare against OpenVLA, a state-of-the-art language-conditioned generalist robot policy (Kim et al., 2024). We note that OpenVLA is trained on a dataset much larger than Bridge, and has roughly $100\times$ more parameters than our policy. While this gives it an advantage in a certain sense, we nonetheless include it as a representative example of a generalist robot policy, and to evaluate the ability of generalist policies to "explore". In particular, we seek to understand whether OpenVLA, deployed with standard prompting, is able to solve the goal task in settings where the goal task is ambiguous given the prompt, and exploration over possible solutions is required (please see Section B.3 for additional discussion on how we prompt OpenVLA). For each task and method, we evaluate on 8 5-episode trials.

**Results.** The results for our real-world experiment are given

in Figure 9, where we plot the average number of successful trials, and average number of attempts to success (in cases where the policy did not interact with both objects in 5 attempts, we count this as 6 attempts). We see that our approach succeeds roughly 40% more than BC or OpenVLA, and requires on average 2 fewer attempts to succeed. As in the Libero experiments, this illustrates the ability of BE to focus its exploration over semantically meaningful features—rather than exploring the entire space of robot configurations, it explores only over the types of reaching and grasping behaviors present in the demonstration dataset—and demonstrates the scalability of our approach to large-scale, real-world settings.

# 6. Discussion

In this work we have proposed behavioral exploration, a novel approach to training policies that explore over the space of behaviors contained in a demonstration dataset. BE requires only a small modification to the standard BC objective to train, and our experimental results demonstrate that BE leads to substantial gains over both standard BC as well as RL-based exploration approaches. We believe our work opens several directions for future work:

- While BE explores over the behaviors present in the demonstration data, there may exist scenarios where the exploration necessary to learn to solve a task requires going *beyond* the space of demonstration data. How can we best utilize the prior information encoded in the demonstrations to speed up exploration in such settings?

- Much attention has been devoted to investigating the ability of LLMs to explore, with mixed results (Krishnamurthy et al., 2024; Nie et al., 2024; Monea et al., 2024). Could BE be combined with LLMs—for example, by finetuning an LLM on the BE objective—to enable effective exploration in language-based domains?

- Our WidowX results demonstrate that BE effectively scales to real-world settings. Scaling this further and combining BE with state-of-the-art generalist robot policies (Kim et al., 2024; Black et al., 2024; Bjorck et al., 2025) is of great interest.

## Acknowledgements

The authors would like to thank the members of RAIL Lab and Max Simchowitz for helpful conversations. This research was partly supported by the RAI Institute, ONR N00014-22-1-2773 and N00014-25-1-2060, and NSF IIS-2150826. This research used the Savio computational cluster resource provided by the Berkeley Research Computing program at UC Berkeley.

## Impact Statement

Exploration can potentially lead to unsafe behaviors—by incentivizing an agent to attempt new behaviors, it is possible some of the behaviors attempted will lead to unsafe actions. While we believe our approach should mitigate this as compared to standard exploration approaches, care should nonetheless be taken. More broadly, advancing agentic systems may have a variety of societal applications. However, we do not feel we need to highlight any in particular here.

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

# A. Proof of Proposition 4.2

For the purposes of the following result, define $w^{\pi_\beta}(s) := \Pr^{\pi_\beta}[s_K = s]$ and $\Omega := \{s : w^{\pi_\beta}(s) > 0\}$. Let $C_\beta := \{i : \exists s \text{ with } w^{\pi_\beta}(s) > 0 \text{ and } \phi(s) = e_i\}$, and $n_\beta := |C_\beta|$ denote the number of directions in feature space $\pi_\beta$ spans at step $K$. We then have the following.

**Proposition A.1** (Full version of Proposition 4.2). *Assume that we are in a deterministic environment and that our feature is a one-hot encoding and only non-zero at the final state. Furthermore, assume that $|\mathcal{A}| < \infty$, that each episode starts from state $s_0$, that we only consider $n_\beta$ rounds of interaction, and take $\epsilon < \min_{s \in \Omega} w^{\pi_\beta}(s)$. Let $h_t$ denote the set of states visited in episodes $1, \ldots, t-1$, and let $\widetilde{\pi}$ denote the policy which at episode $k$ plays actions according to*

$$P^{\pi_\beta}[\,\cdot\mid s, h_t, \operatorname{cov}(h_t \cup \tau) = \max_{\tau'} \operatorname{cov}(h_t \cup \tau)].$$

*Then $\widetilde{\pi}$ is an optimal solution to Objective 4.1.*

*Proof.* Note that in our setting we have that the optimal solution has $\Pr^{\pi}[\phi(s_K) = i] = 1/n_\beta$ for each $i \in C_\beta$ (and does not depend on the visitation to non-terminal states since they all affect coverage equally). Thus, it suffices to show that the policy $\widetilde{\pi}$ in the result statement satisfies $\Pr^{\widetilde{\pi}}[\phi(s_K) = i] = 1/n_\beta$ after playing for $n_\beta$ episodes.

We will complete this proof by induction, where we show that at each episode $t$, we reach a previously unreached terminal state in $C_\beta$. Define $h_t$ denote the set of states visited at episodes $1, \ldots, t-1$, and let $\bar{C}_\beta^t := C_\beta \backslash \{i : \exists s \in h \text{ with } w^{\pi_\beta}(s) > 0 \text{ and } \phi(s) = e_i\}$. It suffices then to show that $|\bar{C}_\beta^t| = n_\beta - t + 1$ for each $t$. The base case, at $t = 1$, is trivial, since $\bar{C}_\beta^1 = C_\beta$.

Now assume that $|\bar{C}_\beta^t| = n_\beta - t + 1$ for some $k < n_\beta$, and we wish to show that $|\bar{C}_\beta^{t+1}| = n_\beta - t$. By construction, there exists some sequence of actions that will take us from $s_0$ to a state $s_K$ with $\phi(s_K) \in \bar{C}_\beta^t$. If we can show that we will reach such a state at step $H$ of episode $k$, the inductive step follows.

To show this, we introduce a second inductive step. In particular, we seek to show that for each step $h$ of episode $k$, we will be in a state $s_k$ for which a path exists to some $s_K$ playing only actions in the support of $\pi_\beta$, and with $\phi(s_K) \in \bar{C}_\beta^t$. The base case is trivial and is true by assumption, as stated above. For the inductive case, assume that we are in such a state at $s_{k-1}$. Then we want to show that we will play an action which will cause us to transition to such a state at $s_k$. As we condition on the event that $\{\operatorname{cov}(h \cup \{s_K\}) = \max_s \operatorname{cov}(h \cup \{s\})\}$, then $\widetilde{\pi}$ by definition will play an action that will allow us to reach some state for which $\phi(s_K) \neq \phi(s)$ for any $s \in h_t$ (we note that such an action in the support of $\pi_\beta$ exists at step $h - 1$ by the inductive assumption). Note also that $\widetilde{\pi}$ will only play actions in the support of $\pi_\beta$ at $s_{k-1}$ and, furthermore, that since we have played $\widetilde{\pi}$ up to step $h - 1$, we have always played actions in the support of $\pi_\beta$, and therefore $w^{\pi_\beta}(s_{k-1}) > 0$.

Let $\bar{\mathcal{A}}(s_{k-1})$ denote the set of actions that would lead to a state $s_k$ from which it is not possible to get to a terminal state in the set $\{s : \phi(s) \in \bar{C}_\beta^t\}$ by playing actions in the support of $\pi_\beta$. For notational convenience, let $\mathcal{E}$ denote the event $\operatorname{cov}(h_t \cup \{s_K\}) = \max_s \operatorname{cov}(h_t \cup \{s\})$. Then if we can show that

$$P^{\pi_\beta}[a \in \bar{\mathcal{A}}(s_{k-1}) \mid s_{k-1}, h_t, \mathcal{E}] = 0 \tag{5}$$

we are done. By the definition of conditional probability, we have

$$P^{\pi_\beta}[a \in \bar{\mathcal{A}}(s_{k-1}) \mid s_{k-1}, h_t, \mathcal{E}] = \frac{P^{\pi_\beta}[\{a \in \bar{\mathcal{A}}(s_{k-1})\} \cap \mathcal{E} \mid s_{k-1}, h_t]}{P^{\pi_\beta}[\mathcal{E} \mid s_{k-1}, h_t]}.$$

Note that

$$P^{\pi_\beta}[\{a \in \bar{\mathcal{A}}(s_{k-1})\} \cap \mathcal{E} \mid s_{k-1}, h_t] = P^{\pi_\beta}[\{a \in \bar{\mathcal{A}}(s_{k-1})\} \cap \mathcal{E} \cap \{a_h, \ldots, a_{H-1} \in \operatorname{support}(\pi_\beta)\} \mid s_{k-1}, h_t]$$

by definition of $P^{\pi_\beta}$. Furthermore, note that the event

$$\{a \in \bar{\mathcal{A}}(s_{k-1})\} \cap \mathcal{E} \cap \{a_h, \ldots, a_{H-1} \in \operatorname{support}(\pi_\beta)\}$$

implies the event

$$\{\phi(s_K) \notin C_\beta, s_K \notin h_t\}$$

since $a \in \bar{\mathcal{A}}(s_{k-1})$ cannot lead to any $s_K$ with $\phi(s_K) \in \bar{C}_\beta$, by definition, if we only take actions in the support of $\pi_\beta$, but since we condition on the event $\mathcal{E}$ we must reach a terminal state not in $\boldsymbol{h}_t$. This implies that the above can be bounded as

$$\leq \frac{P^{\pi_\beta}[\{\phi(s_K) \notin C_\beta, s_K \notin \boldsymbol{h}_t\} \mid s_{k-1}, \boldsymbol{h}_t]}{P^{\pi_\beta}[\mathcal{E} \mid s_{k-1}, \boldsymbol{h}_t]}.$$

However, since $w^{\pi_\beta}(s_{k-1}) > 0$, the final state reached by playing $\pi_\beta$ from this state must satisfy $\phi(s_K) \in C_\beta$. Thus, we have $P^{\pi_\beta}[\{\phi(s_K) \notin C_\beta, s_K \notin \boldsymbol{h}_t\} \mid s_{k-1}, \boldsymbol{h}_t] = 0$. Furthermore, by the inductive assumption we have $P^{\pi_\beta}[\mathcal{E} \mid s_{k-1}, \boldsymbol{h}_t] > 0$. Together this proves that (5) holds, which proves the inner inductive step.

The inner induction then immediately implies the outer induction step, that we reach a terminal state at episode $k$ in $\bar{C}_\beta^t$, so that $|\bar{C}_\beta^{t+1}| = n_\beta - t$. Repeating this for each $k \leq n_\beta$, we have that $\widetilde{\pi}$ will visit terminal states $\{s_K^t\}$ such that $\{i : \exists k, \phi(s_K^t) = \boldsymbol{e}_i\} = C_\beta$, which implies that $\Pr^{\widetilde{\pi}}[\phi(s_K) = i] = 1/n_\beta$ after playing for $n_\beta$ episodes, thus proving the result. $\qquad \square$

# B. Experiment Details

We make several notes on hyperparameters for all experiments. For all experiments, for both BE and BC, we use the diffusion policy architecture proposed by Dasari et al. (2024) and utilize their code base as the starting point for our method. We experiment with several different parameter settings for architecture size, and for all BC evaluations run the same parameter sweep as we consider for BE, reporting results for the best-performing architectures. For all experiments, we use action chunking, where the model predicts several future actions (in evaluation, we roll out all actions before recomputing a new set of actions)—we specify action chunk length used for each setting below. For BE, in addition to standard hyperparameters common with BC, it also requires specifying the context length (history length), and future trajectory length (in settings where $\tau^t$ is very long, how far in the future we should consider when calculating coverage). In deployment, in cases where the context length is shorter than the history we have collected, we randomly downsample the history. For all experiments we use the definition of $\text{cov}(\cdot)$ stated in the main text—we specify what we choose for $\phi(\cdot)$ in each case below. All BC and BE models were trained with the following hyperparameters in common—we specify other hyperparameters in the following.

Table 1: Common hyperparameters for all BE and BC experiments.

| Hyperparameter | Value |
|----------------|-------|
| Learning rate | 3e-4 |
| LR scheduler | cosine |
| Warmup steps | 2000 |

## B.1. D4RL Antmaze and Kitchen Experiments

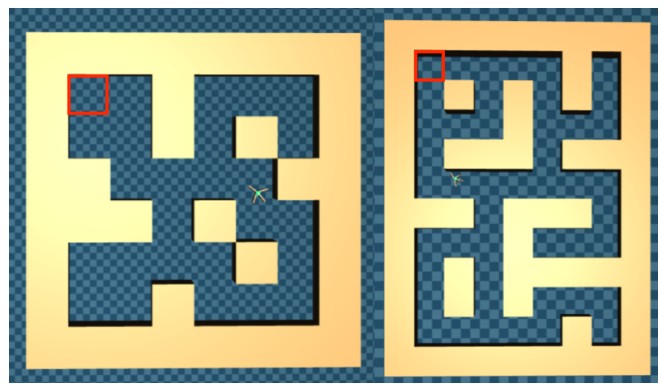

Figure 12: Visualization of D4RL Antmaze `medium` and `large` environments. Red squares denote a single "region", used to calculate the region count in Figure 4. Please see Figure 2(a) of Wilcoxson et al. (2024) for goal locations.

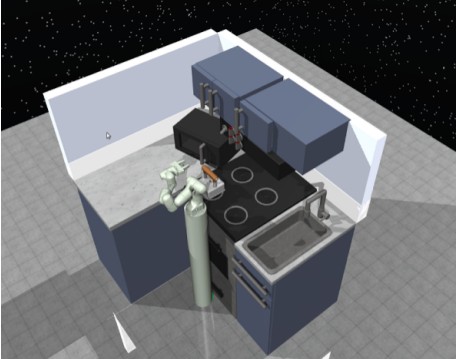

Figure 13: Visualization of D4RL Kitchen environment

We utilize the default D4RL environments and datasets in all respects except for the episode length of Antmaze: for Antmaze `medium` we shorten the episode length from 1000 to 500, and for Antmaze `Large` we shorten from 1000 to 750. The shorter horizon better tests the ability of each approach to realize directed exploration; with a longer horizon randomly exploring over the maze is itself an affective strategy. As stated in the text, we evaluate based on the number of goals reached (for Antmaze) or tasks completed (for Kitchen). For Antmaze, we utilize the goal locations given in Wilcoxson et al. (2024), and count the goal as reached if the agent reaches with a distance of 0.5 of a square of side length 1.5 centered at the goal. For Kitchen, we utilize the environments default success detector to determine whether a goal is reached. We do not require that these are completed in the same episode, simply over the entire sequence of episodes (so over 20k steps for Antmaze and 3k steps for Kitchen).

For all RL baselines, we use the implementations and pretraining checkpoints provided by Wilcoxson et al. (2024). We

utilize the default hyperparameters for each method as given in Wilcoxson et al. (2024) with the exception of the length of the online warmup phase before RL training start, given that our time horizon is shorter than is used in Wilcoxson et al. (2024). For the warmup length, for each method in each environment we sweep over several different values, and choose the value that leads to the best final performance.

For BE, we utilize the definition for coverage given in the main text, and as a feature vector $\phi(s)$, feed the state into a randomly initialized neural network, taking the last layer as the feature vector. In all cases we use a 3-layer fully connected network with hidden dimension 128 and output dimension 32 (and set $\lambda = 0.01$ for our regularization). In deployment, we set the history to be a randomly selected subset of past states visited (which is resampled at the start of each episode).

Figure 3 and Figure 4 gives the performance averaged across all goals and both Antmaze `medium` and `large`, and Table 2 provides the final values achieved in each plot. Please see Figure 14-19 for individual results.

Table 2: Final success rate from Figure 7 and 8.

|  | BE | BC | SUPE | HILP | ExPLORe | RND |
|---|---|---|---|---|---|---|
| Antmaze (goals) | **0.556** $\pm$ 0.013 | 0.285 $\pm$ 0.015 | 0.195 $\pm$ 0.013 | 0.128 $\pm$ 0.012 | 0 $\pm$ 0 | 0 $\pm$ 0 |
| Antmaze (regions) | **24.118** $\pm$ 0.253 | 20.056 $\pm$ 0.150 | 16.963 $\pm$ 0.258 | 18.934 $\pm$ 0.218 | 3.606 $\pm$ 0.040 | 3.475 $\pm$ 0.044 |
| Kitchen | 0.922 $\pm$ 0.014 | 0.306 $\pm$ 0.017 | **0.956** $\pm$ 0.011 | 0.913 $\pm$ 0.015 | 0.075 $\pm$ 0.013 | 0.081 $\pm$ 0.013 |

Table 3: Hyperparameters for D4RL BE experiments.

| **Hyperparameter** | Antmaze Medium | Antmaze Large | Kitchen |
|---|---|---|---|
| Batch size | 256 | 256 | 256 |
| Action chunk size | 10 | 10 | 5 |
| Hidden size | 256 | 128 | 64 |
| Number of Heads | 8 | 4 | 1 |
| Number of Layers | 4 | 4 | 3 |
| Feedforward dimension | 256 | 512 | 512 |
| Coverage future length | 200 | 100 | 50 |
| Context history length | 100 | 50 | 50 |

Table 4: Hyperparameters for D4RL BC experiments.

| **Hyperparameter** | Antmaze Medium & Large | Kitchen |
|---|---|---|
| Batch size | 256 | 256 |
| Action chunk size | 10 | 5 |
| Hidden size | 256 | 64 |
| Number of Heads | 8 | 1 |
| Number of Layers | 4 | 3 |
| Feedforward dimension | 512 | 128 |

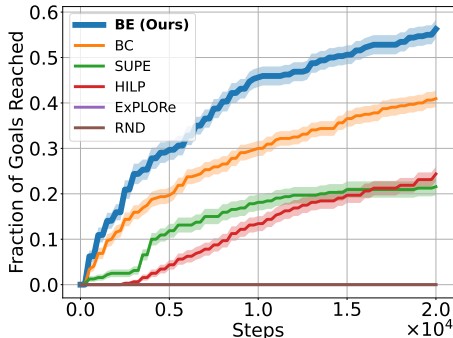

Figure 14: Performance on Antmaze `medium`, measuring number of goals reached.

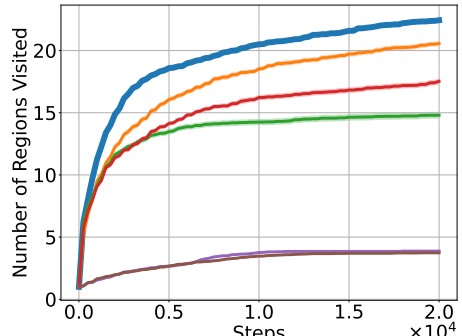

Figure 15: Performance on Antmaze `medium`, measuring number of regions reached.

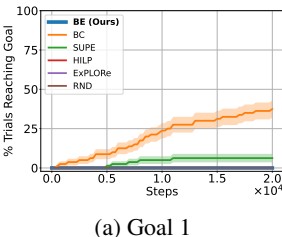

(a) Goal 1

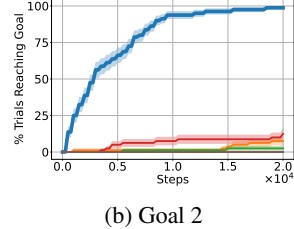

(b) Goal 2

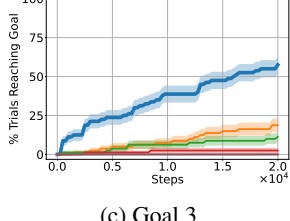

(c) Goal 3

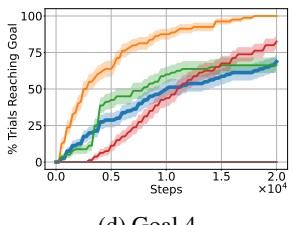

(d) Goal 4

Figure 16: Performance on individual goals on Antmaze `medium`.

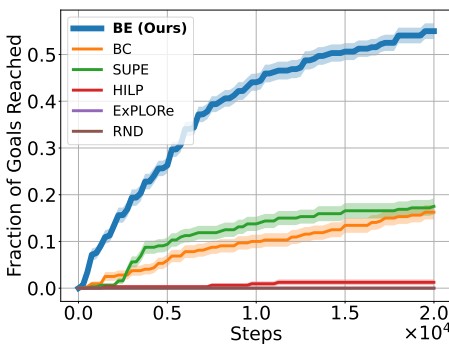

Figure 17: Performance on Antmaze `large`, measuring number of goals reached.

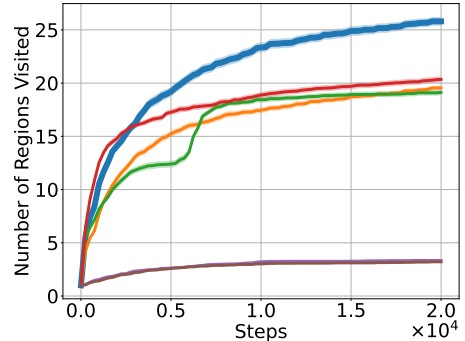

Figure 18: Performance on Antmaze `large`, measuring number of regions reached.

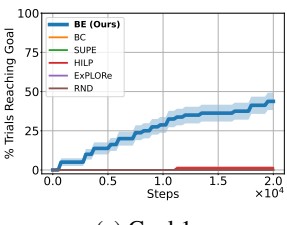

(a) Goal 1

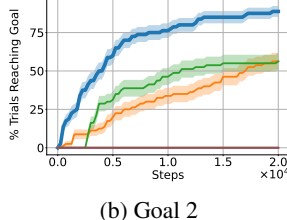

(b) Goal 2

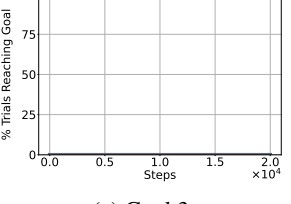

(c) Goal 3

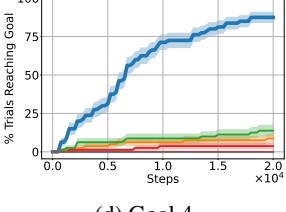

(d) Goal 4

Figure 19: Performance on individual goals on Antmaze `large`.

## B.2. Libero Experiments

Table 5: Libero tasks for scenes in Figure 10.

| Scene | Task Commands |
|---|---|
| Upper Left | pick up the alphabet soup and put it in the tray |
| | pick up the butter and put it in the tray |
| | pick up the cream cheese and put it in the tray |
| | pick up the ketchup and put it in the tray |
| | pick up the tomato sauce and put it in the tray |
| Upper Right | close the top drawer of the cabinet |
| | put the black bowl in the top drawer of the cabinet |
| | put the black bowl on the plate |
| | put the black bowl on top of the cabinet |
| | put the ketchup in the top drawer of the cabinet |
| Lower Left | put the frying pan on the stove |
| | put the moka pot on the stove |
| | turn on the stove |
| | turn on the stove and put the frying pan on it |
| Lower Right | pick up the book and place it in the front compartment of the caddy |
| | pick up the book and place it in the left compartment of the caddy |
| | pick up the book and place it in the right compartment of the caddy |
| | pick up the red mug and place it to the right of the caddy |
| | pick up the white mug and place it to the right of the caddy |

For the hidden-task evaluation, for both our approach and BC we train the model without task conditioning, but we do provide a scene conditioning vector for each (a one-hot 21-dimensional vector indicating which of the 21 scenes the agent is operating in). For each task, we run with a horizon of 300 steps, and utilize the environment's built-in success detector. We condition all methods on the timestep, proprioceptive state, and both camera views (using a ResNet-18 encoder to encode images). For the task-conditioned evaluation, we use a one-hot encoding for each task. We set $\phi(s) = \cos(As_{\mathrm{proprio}} + b)$, where $s_{\mathrm{proprio}} \in \mathbb{R}^9$ denotes the proprioceptive state, and $A \in \mathbb{R}^{16 \times 9}$ and $b \in \mathbb{R}^{16}$ are randomly initialized parameters. At deployment, for BE, we sample a single state from each of the last three episodes and repeat them each for 1/3 of the total history length.

For BE and BC, we swept over the stated hyperparameters, and include the results for the best-performing ones. For SUPE we utilize the default hyperparameters for the vision-based experiments from Wilcoxson et al. (2024). For BC with action noise, at every step we sample noise $w \sim \mathcal{N}(0, 0.15 \cdot I)$ which we add to the action computed by the BC policy. For BC with history conditioning, we utilize a history length of 100 (the same as that utilized by BE) and otherwise use the same hyperparameters as standard BC.

Figure 7 and 8 provide results averaged across all 90 tasks, and we state the final success rates for these evaluations in Table 6. Please see Figure 20-23 for individualized results.

Table 6: Final success rate from Figure 7 and 8.

| | BE | BC | BC (Action Noise) | BC (History) | BC (Random Task) | SUPE |
|---|---|---|---|---|---|---|
| Task hidden | **0.824** $\pm$ 0.010 | 0.590 $\pm$ 0.013 | 0.684 $\pm$ 0.013 | 0.643 $\pm$ 0.013 | 0.359 $\pm$ 0.013 | 0.267 $\pm$ 0.047 |
| Task provided | **0.993** $\pm$ 0.002 | 0.977 $\pm$ 0.004 | 0.950 $\pm$ 0.006 | 0.804 $\pm$ 0.011 | - | - |

Table 7: Hyperparameters for Libero BE and BC models.

| Hyperparameter | BE | BC |
|---|---|---|
| Batch size | 150 | 150 |
| Action chunk size | 20 | 10 |
| Image encoder | ResNet-34 | ResNet-34 |
| Hidden size | 256 | 256 |
| Number of Heads | 8 | 8 |
| Number of Layers | 4 | 4 |
| Feedforward dimension | 512 | 512 |
| Coverage future length | 200 | - |
| Context history length | 100 | - |

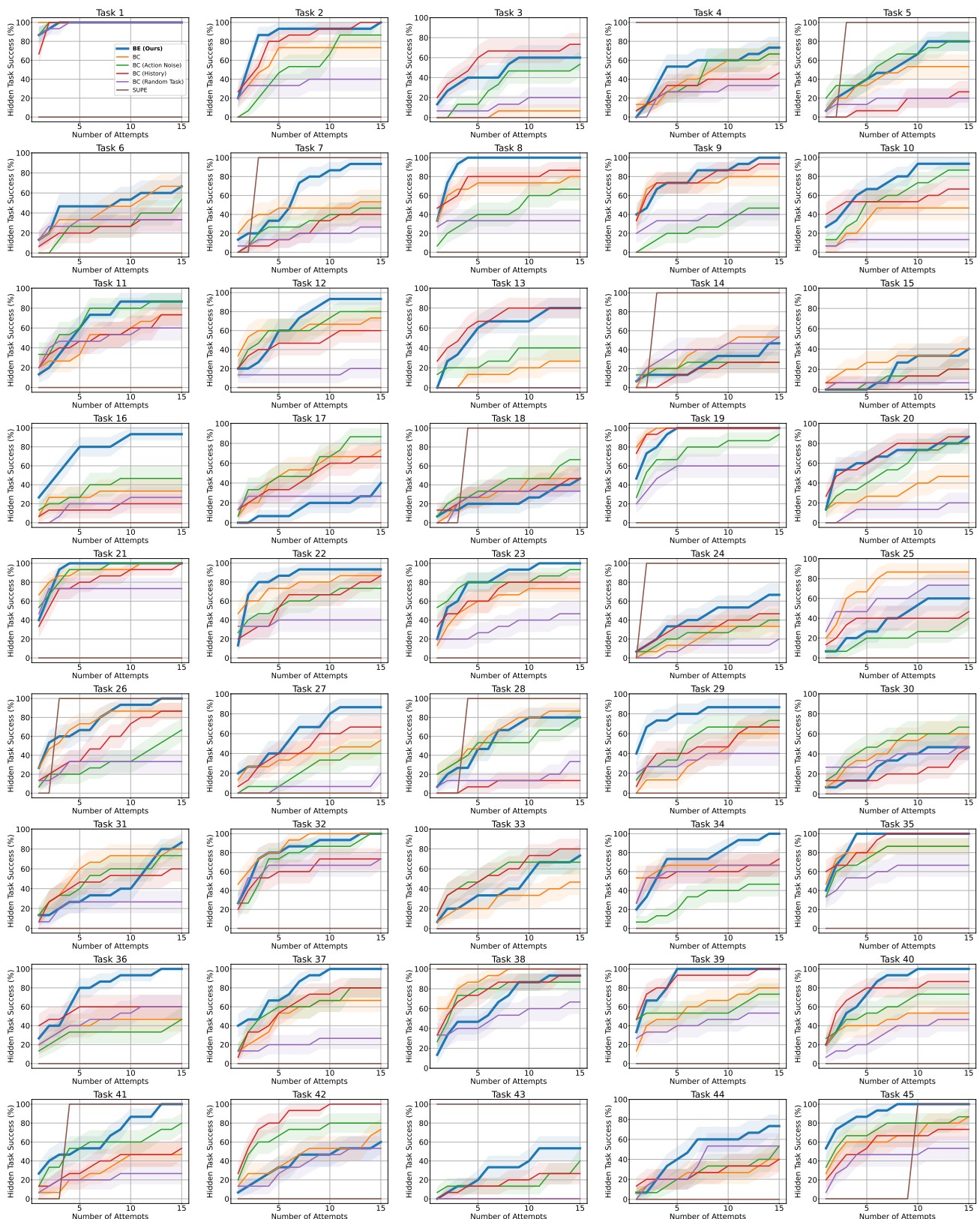

Figure 20: Performance on individual Libero tasks (1-45) in evaluation with task hidden.

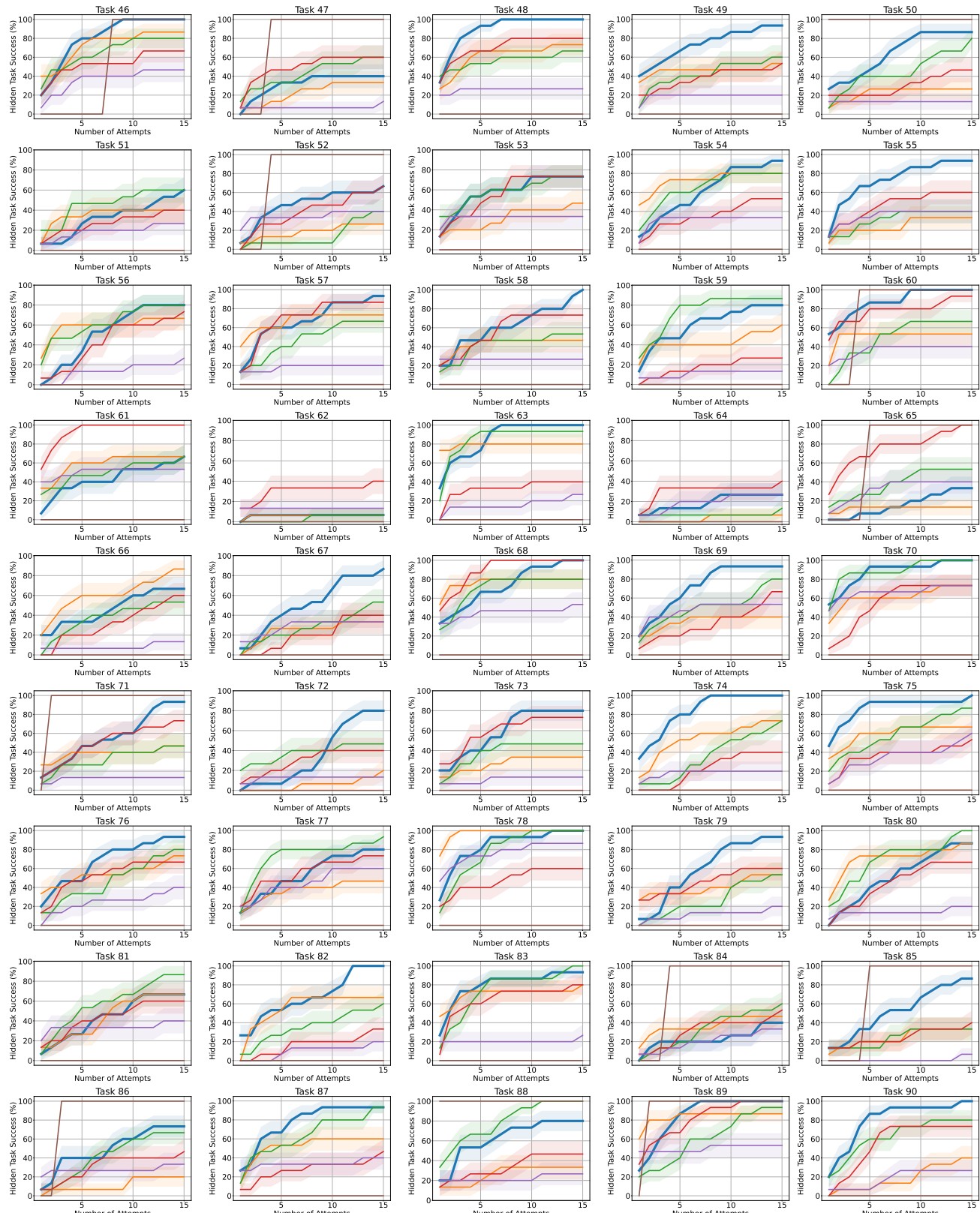

Figure 21: Performance on individual Libero tasks (46-90) in evaluation with task hidden.

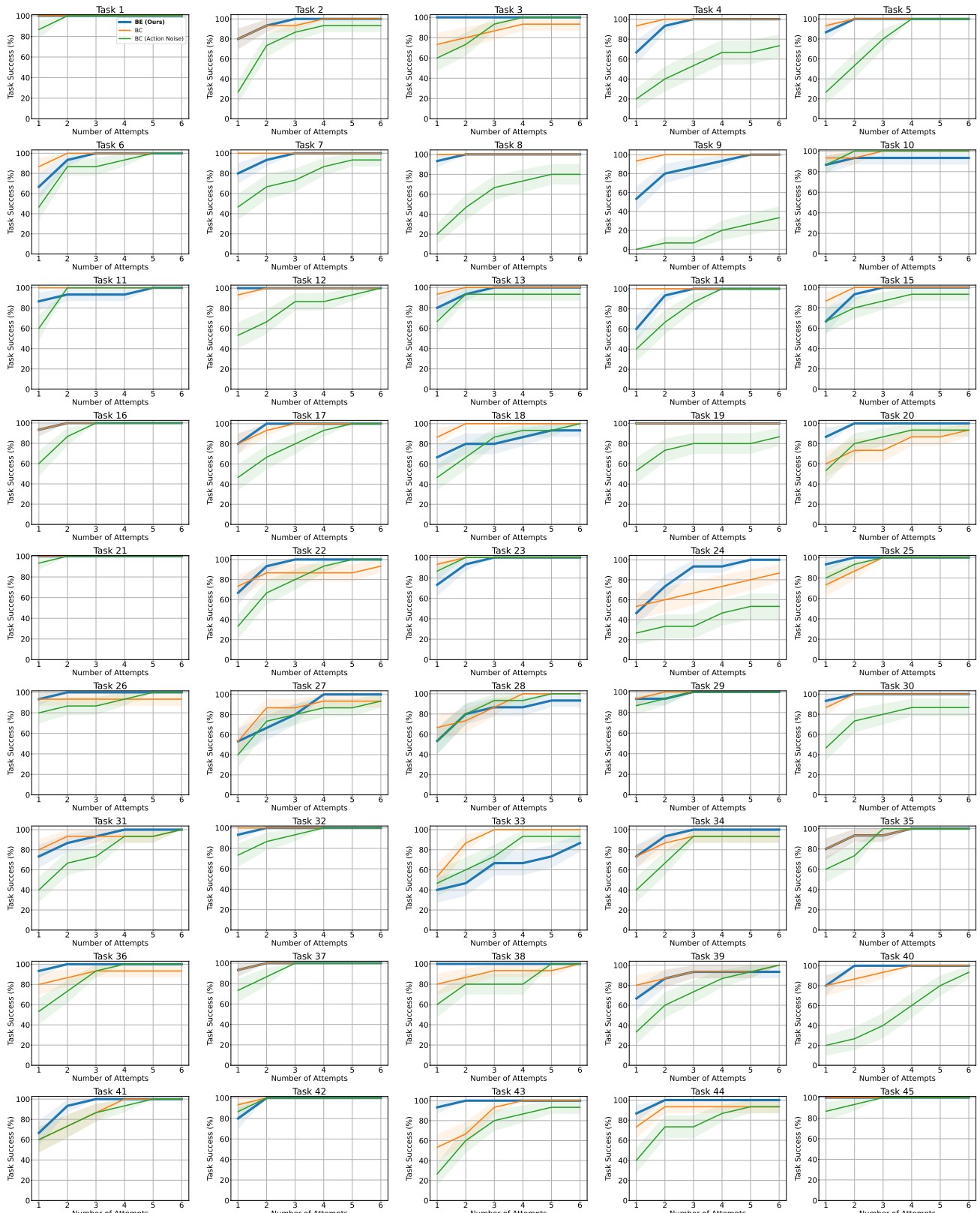

Figure 22: Performance on individual Libero tasks (1-45) in evaluation with task provided.

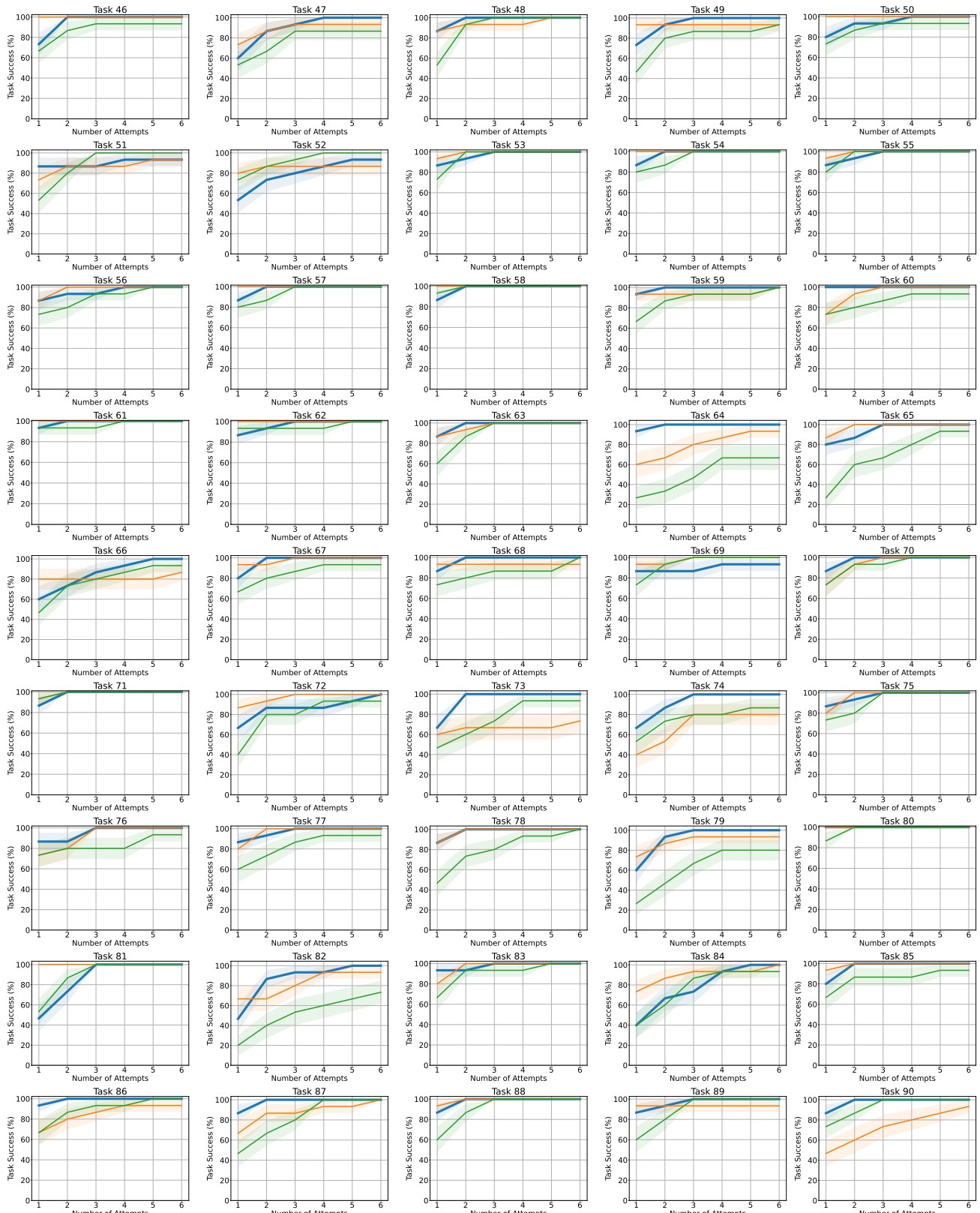

Figure 23: Performance on individual Libero tasks (46-90) in evaluation with task provided.

## B.3. WidowX Experiments

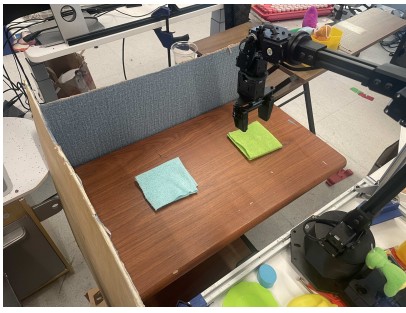

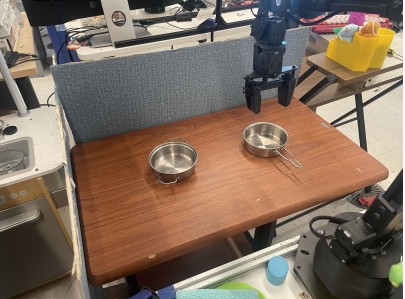

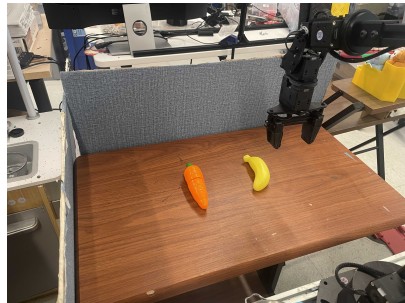

Figure 24: WidowX Task 1.     Figure 25: WidowX Task 2.     Figure 26: WidowX Task 3.

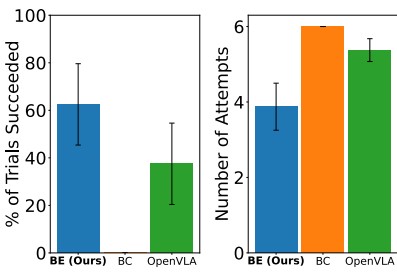

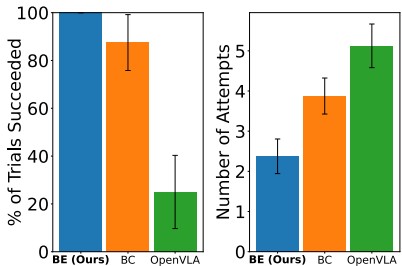

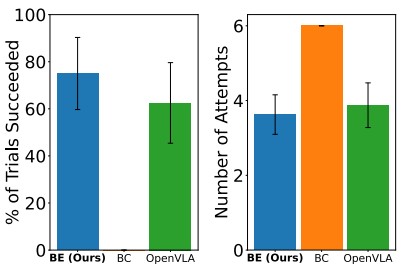

Figure 27: Results on WidowX Task 1.     Figure 28: Results on WidowX Task 2.     Figure 29: Results on WidowX Task 3.

Individual results on each WidowX task are given in Section B.3. We note the BC fails to succeed a single time on 2/3 tasks, and that our approach achieves the highest success rate and lowest average number of attempts on all tasks.

We use delta end-effector control with a frequency of 5 Hz. We use an RGB camera to capture the top-down third-person view of the robot workspace, and use 128×128 images as input to our control policy.

While the Bridge dataset contains text labels for the demonstrations, since we are evaluating the ability to select diverse behaviors, for both the BC baseline and BE we do not use text conditioning. We use the same form of $\phi(s)$ as with Libero, but instead of conditioning on the full proprioceptive state we condition only on the relative $(x, y, z)$ coordinates (that is, if we are at state $s'$, then we define $\phi(s; s') = \cos(A(s_{xyz} - s'_{xyz}) + b)$.

We evaluate on a reaching task, as stated in the main text. We consider a reach to be successful if the end-effector makes contact with the object.

As OpenVLA requires language conditioning, we chose objects in the same scene that were relatively similar in order to allow them both to correspond to the same language command. For Task 1, we use the command "pick up the cloth", for Task 2 "pick up the silver pot", and for Task 3 "pick up the object". We note that OpenVLA is not able to condition on a history of past observations and, as such, at deployment, each episode is effectively independent from past episodes. Our aim in this experiment is therefore to test whether OpenVLA tends to focus its behavior on a single possible solution, or if it attempts diverse behaviors over multiple episodes.

We also experiment with modifying the prompt for OpenVLA to induce more diverse behaviors. In particular, on Figure 25, we experiment with prompting OpenVLA with either "pick up the silver pot" vs "pick up the other silver pot". We hypothesize that by adding the word "other" to the prompt, this may encourage OpenVLA to attempt to pick up the pot that it is initially biased away from. We provide the results of our evaluation on these prompts in Table 8, where we see that OpenVLA exhibits a strong bias to the right pot and, while changing the prompt does cause OpenVLA to reach for the left more frequently, it still exhibits a strong rightward bias, and the prompt modification also causes it to fail to reach any pot more frequently. We hypothesize that this may be due to the modified command being more out-of-distribution for

OpenVLA, hurting its overall performance. These results illustrate that, while modifying the prompt may help somewhat in encouraging OpenVLA to explore, it is far from exhibiting robust exploratory behavior.

Table 8: Performance of OpenVLA with different prompts. We consider running 8 rollouts on the task illustrated in Figure 25, and indicate in how many episodes the end effector made contact with the left, right, or neither pot.

| Command | Left | Right | Neither |
|---|---|---|---|
| "pick up the silver pot" | 1/8 | 7/8 | 0/8 |
| "pick up the other silver pot" | 2/8 | 4/8 | 2/8 |

Table 9: Hyperparameters for WidowX BE and BC models.

| Hyperparameter | BE | BC |
|---|---|---|
| Batch size | 2560 | 1200 |
| Action chunk size | 4 | 4 |
| Image encoder | ResNet-34 | ResNet-34 |
| Hidden size | 512 | 512 |
| Number of Heads | 8 | 8 |
| Number of Layers | 6 | 6 |
| Feedforward dimension | 2048 | 2048 |
| Coverage future length | 20 | - |
| Context history length | 50 | - |

## B.4. Understanding Behavioral Exploration

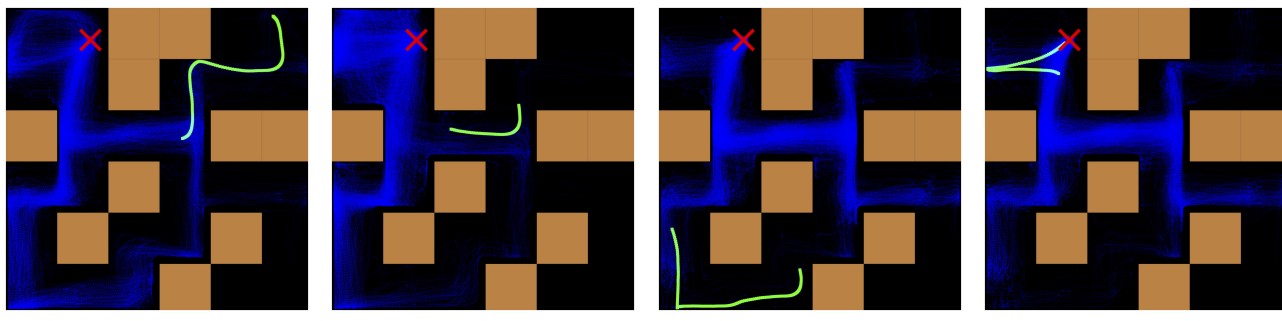

Figure 30: BE trajectory rollouts (blue) when the BE policy is conditioned on a particular history segment (green). Red "x" denotes the starting point for the BE rollouts. In all cases BE adapts its behavior to explore the regions not covered by the conditioning trajectory.

Finally, we seek to provide some insight into the behavior of BE, in particular to what extent BE adapts to the history conditioning, and how critical that adaptation is for effective exploration. We consider here the Maze2D medium environment from D4RL.

We first consider BE's ability to adapt to its history conditioning. In Figure 30, we plot the trajectories induced by BE when it is conditioned on a history of a single trajectory, shown in green, and a value of exp corresponding to high future coverage. With this conditioning we would expect BE to visit *different* states than those in the history. We see that this is indeed the case—in the first two examples, where the conditioning trajectory is on the right side of the maze, the trajectories induced by BE concentrate on the left side, while in the final two examples where the conditioning trajectory is on the top or bottom, the trajectories induced by BE concentrate in the central part of the maze. This illustrates the adaptivity of BE to its history, and BE's ability to take actions leading to the collection of observations that are novel relative to its history.

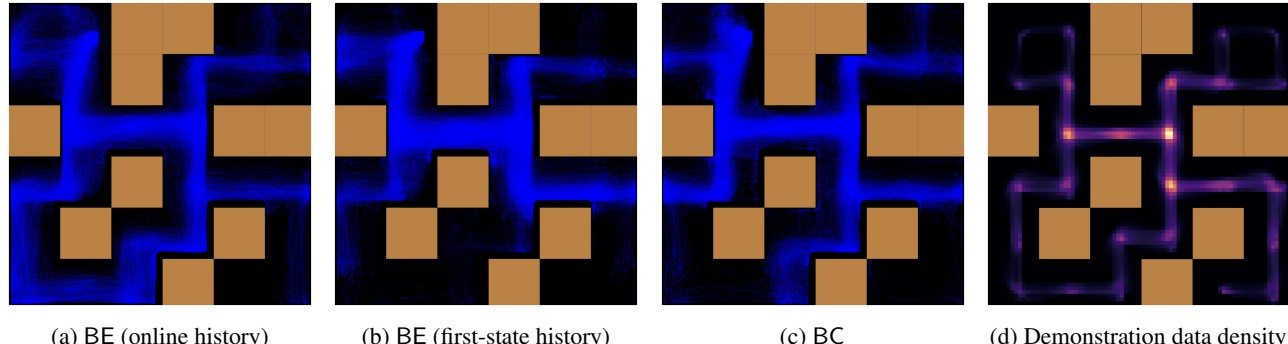

(a) BE (online history)     (b) BE (first-state history)     (c) BC     (d) Demonstration data density

Figure 31: Comparison of BE conditioned with history of past online observations, BE conditioned on only the first state, and BC. We see that conditioning on the history of past states visited is critical to achieving high coverage, and that this induces coverage over low-density regions in the demonstration data that BC fails to reach.

We next seek to understand how critical this history adaptation is to overall exploration behavior. Figure 31a shows rollouts from BE in its standard operation, where we condition it on a history sampled from the past states it has visited in online rollouts. Figure 31b shows rollouts from BE but where, instead of conditioning it on its history of past state, we simply condition it on a history that only contains the first state visited at the start of rollouts. We see that the coverage is significantly greater in Figure 31a than Figure 31b—while trajectories in Figure 31b primarily visit points in the central part of the maze, in Figure 31a BE achieves coverage throughout the maze. This highlights the role of adaptivity in inducing effective exploration—by adapting to its history, BE is able to achieve much more diverse coverage than if it only considers a fixed history, and much more diverse coverage than BC (Figure 31c). Furthermore, if we compare the coverage achieved by BE vs BC relative to the demonstration data coverage density (Figure 31d), we see that BC primarily visits the high-density regions in the training data—particularly through the central part of the maze—while BE achieves significantly higher coverage in low-density regions in the training data—particularly the lower left corner. This illustrates BE's ability to increase coverage in the low-density regions in the demonstration data.

