# OpenReview forum: "Behavioral Exploration: Learning to Explore via In-Context Adaptation"
_ICML.cc/2025/Conference — ICML 2025 poster_

### Official Review · Reviewer_RYNM · 2025-02-21

**Overall Recommendation:** 2

**Summary:**

This paper presents a method to predict expert actions based on past observations and predict how "exploratory" the expert's behaviors are relative to the context. This design enables the model to mimic the behavior of an expert, and select the most relative experiences to explore.

---

Update after rebuttal: Many important results, such as tabular results, have not been provided. The supplemented videos lack descriptions and comparisons. This paper still needs more polish. The reviewer's score is unchanged.

**Claims And Evidence:**

1. The claim in the introduction is that existing works are "slow and expensive." This is not supported by any evidence or citations.
2. The introduction example claims that the robots should not explore "pick up a plate or pick up the same cup." The reviewer doesn't agree with this because exploring (although not successful attempts) can be quite beneficial. Exploring wrong experiences can be quite beneficial in avoiding erroneous actions. For example, exploring the action of picking up a plate can help avoid making such a mistake.

**Essential References Not Discussed:**

The literature review is reasonably extensive.

**Experimental Designs Or Analyses:**

1. Variances are not shown in Figure 2-Figure 4.
2. Tabular results should be provided as well, at least in the appendix.
3. Supplementary videos are recommended.
4. It seems that Figure 2 - Figure 4 do not present consistent results. For example, HILP doesn't always outperform SUPE.

**Methods And Evaluation Criteria:**

1. Measuring coverage through the inverse feature matrix needs better motivation. Compared to similar lines of work in this domain (the coverage scheme) is highly recommended.

2. D4RL is a relatively new benchmark while evaluating other existing standard benchmarks (e.g., Atari or Mujoco) are recommended.

**Other Comments Or Suggestions:**

See above sections.

**Other Strengths And Weaknesses:**

N/A

**Questions For Authors:**

Please answer the questions mentioned above.

**Relation To Broader Scientific Literature:**

The reviewer cannot agree with the motivations behind this work (see the above comments). Therefore, the significance of this work is questionable.

**Theoretical Claims:**

N/A

---

> ### Author Rebuttal · Authors · 2025-04-01
>
> We thank the reviewer for their feedback. We have added videos illustrating the performance of our approach and baselines on our real-world WidowX setting [here](https://drive.google.com/drive/folders/1-B7kgD9lXVR41WyhtR1XuLepQXI8QR7W?usp=share_link), and provide clarification on additional points below.
>
> ### The claim in the introduction is that existing works are "slow and expensive”…
> It is well-known that randomized exploration, on which the majority of practical RL approaches are based, is “expensive” in the sense that it requires many interactions before it discovers the correct behavior (see e.g. [1]). Furthermore, most RL algorithms update their behavior “slowly” by gradually fitting a value function to newly collected data, and updating their policy to maximize this value function. While this approach can fit the data after sufficiently many gradient steps, to avoid training instability one must typically limit the number of gradient steps per sample, causing the policy to lag behind the behavior that could in principle be learned from the data (see e.g. [2]). We believe these statements are backed up by our experiments (Figures 2 and 3, in particular), which illustrate that our approach of in-context adaptation leads to much faster exploration and task-solving ability than RL approaches, and furthermore these observations are common throughout the RL literature.
>
> ### The introduction example claims that the robots should not explore…
> While we concur that in certain settings—in particular, when very little prior knowledge is available—taking “wrong” actions is indeed useful, in settings where one already knows an action is “wrong” taking this action is of limited usefulness. For example, if the agent knows what a “cup” is, then it already knows that picking up a plate is incorrect, so attempting to pick up the plate does not help it learn anything new about which cup it should pick up. Furthermore, as there are many possible "wrong" behaviors, exploring them all is very sample inefficient.
>
> The focus of this paper is on scenarios where the prior knowledge (in our case, in the expert demonstrations) is rich enough to avoid many “wrong” behaviors, and where we want to quickly explore over potentially correct behaviors. We believe this is useful in many real-world settings. For example, ideally we would want a robot policy that follows instructions (when we tell it to “pick up the pot” it picks up a pot, and not some other object) but that can try different behaviors until it picks up the correct pot. Our experiments illustrate that OpenVLA, a state-of-the-art generalist robot policy, can do the former yet not the latter, and existing work on exploration can achieve the latter (after many samples attempting every possible behavior) but not the former—our work fills a gap in the literature by enabling a policy to achieve both simultaneously.
>
> ### D4RL is a relatively new benchmark…
> We would like to emphasize that D4RL is nearly 5 years old and has become the standard benchmark for nearly all work in offline RL since its release (see e.g. [3]-[5]). Furthermore, other RL benchmarks (such as Atari) do not have standard offline datasets—which is critical in our setting—and are therefore not amenable to this work.
>
> ### Variances are not shown…
> Figures 2-4 do include error bars—see, for example, the BC curve of Figure 2. As stated in the main text, these error bars denote 1 standard error.
>
> ### Tabular results…
> We will include tabular results in final version of the paper, as requested.
>
> ### Consistent results in Figures 2-4
> Figures 2 and 3 consider different metrics—Figure 2 plots the number of goals reached, while Figure 3 the total number of regions reached—and Figure 4 a different environment entirely, and so we would not necessarily expect the ordering to be the same.
>
> ### Measuring coverage…
> As noted in footnote 1 in our paper, measuring coverage through the inverse feature matrix is a canonical technique in statistics and, furthermore, is common in the RL theory literature (e.g. [6]-[8]).
>
> [1] Dann, Chris, et al. "Guarantees for epsilon-greedy reinforcement learning with function approximation." ICML, 2022.
>
> [2] Li, Qiyang, et al. "Efficient deep reinforcement learning requires regulating overfitting." ICLR, 2023.
>
> [3] Chen, Lili, et al. "Decision transformer: Reinforcement learning via sequence modeling." NeurIPS, 2021.
>
> [4] Kostrikov, Ilya, et al. "Offline reinforcement learning with implicit q-learning." arXiv, 2021.
>
> [5] Kumar, Aviral, et al. "Conservative q-learning for offline reinforcement learning." NeurIPS, 2020.
>
> [6] Jin, Chi, et al. "Provably efficient reinforcement learning with linear function approximation." COLT, 2020.
>
> [7] Wagenmaker, Andrew J., et al. "Reward-free rl is no harder than reward-aware rl in linear markov decision processes." ICML, 2022.
>
> [8] Zhou, Dongruo, et al. "Nearly minimax optimal reinforcement learning for linear mixture markov decision processes." COLT, 2021.

---

> > ### Comment · Reviewer_RYNM · 2025-04-01
> >
> > Good moves. These points will definitely improve your drafts.
> > However, these are major revisions and although the authors promise to make many changes. The reviewer cannot evaluate The modified draft due to the challenging reviewing policies, the reviewer cannot decide whether to change his mind before seeing the  revision. A submission is recommended therefore.

---

> > > ### Author Response · Authors · 2025-04-07
> > >
> > > We would like to thank the reviewer for their follow-up response. We respectfully disagree, however, with the assessment that the proposed changes entail a major revision. The changes we have outlined in response to the reviewer’s initial feedback are primarily clarifications, and we will plan to highlight the same clarifying points in the final version of the paper as we have highlighted in our rebuttal. As these are primarily small expositional changes, we believe they are well within the scope of the standard revisions that occur in the review process at ICML.
> > >
> > > If the reviewer feels that our rebuttal has addressed their concerns, we would greatly appreciate if they would be willing to reconsider their assessment of our work.
> > >
> > > Best Regards,
> > > Authors

---

### Official Review · Reviewer_Qbgd · 2025-03-12

**Overall Recommendation:** 4

**Summary:**

This paper proposes a method—Behavioral Exploration (BE)—that is capable of online in-context task adaptation while learning offline from a set of reward-free expert demonstrations. Such an ability is achieved by conditioning the policy on the coverage measure, which reweights the probabilities of less frequent trajectories while remaining under the distribution of expert policies. This approach is compared with existing exploration algorithms, zero-shot RL methods, and a modern VLA in maze-like and robotic environments, and also on a real-world robotic task. The claimed contributions of the paper are the method’s ability to generalize to new goals faster than baseline approaches and the scalability to both simulated and real-world robotic tasks.

**Claims And Evidence:**

There are three main claims in the paper:

- [C1] The proposed method is able to learn from possibly not optimal RL data to quickly adapt to the task

This claim is supported by experiments on two D4RL benchmarks: AntMaze and Franka Kitchen. The proposed method is seems to be more sample efficient while exploring the AntMaze environment, but acts on par with zero-shot HILP and falls short of SUPE. To my mind, the evidence authors provide is not enough to make conclusions of faster adaptation due to similar performance of the BE with the existing ones.

[Concern 1] At the same time, authors argue that their method is easier to train, compared to unsupervised zero-shot methods. Although I have an intuition this might be true, I find this explanation unrelated to the original claim of fast adaptation. Thus, I find the evidence not convincing enough to support the claim of faster adaptation.

- [C2] The method can learn to explore from diverse robotic demonstrations

To show the method is capable of in-context exploration, authors choose the Libero 90 dataset, which consists of human demonstrations of different tasks in the Libero robotic simulator. Authors employ two approaches; first, they hide the final goal and measure the number of attempts of their method and the baselines to achieve the goal. In the second, the task is provided. For a baseline, the behavior cloning (BC) algorithm is used. In both scenarios, the BE approach finds the goal in fewer attempts compared to BC.

[Concern 2] Although the experimental results are positive, the selection of baselines appears insufficient, as only BC is included. Based on the previous experiment, it would be reasonable to include unsupervised zero-shot RL methods, provided they do not require rewards during training. Without additional experiments, the claim is only weakly supported.

[Concern 3] Besides that, I have a question to the authors. In Appendix B.2 they mention that when the goal is hidden, BC and BE are still provided with “a scene conditioning vector”. What does this vector consist of?

- [C3] The method scales to the real-world environments

To support this clam, authors train their method on the BridgeData V2 dataset and set up a robotic arm WidowX 250 to solve three tasks, where it needs to reach the right object on a table. The trial is counted as a success if the right object was picked at least once in five attempts. Two methods are chosen for the comparison; OpenVLA, the recent Vision-Language-Action model that receives an image and language instruction to produce an action. The second baseline is behavior cloning. It is shown that BE has more successful trials compared to baselines, which indicates it is able to demonstrate exploration behavior.

[Concern 4] While the comparison with OpenVLA provides some insights, its relevance may be questioned. According to Appendix B.3, the language command for VLA was fixed for all trials, whereas BE, being an explorative algorithm, adapts by conditioning on a coverage value to generate new trajectories. In contrast, VLA is designed to follow the language commands commands without feedback from the environment. Thus, after touching one of the object once, VLA does not receive any meaningful signal whether the episode was done successfully or not, which eliminates the incentive to try another object. For instance, it would be interesting to see how the results change if, after an initial unsuccessful trial, the model were prompted to select  “*another”* object. Given these differences, the motivation for the current comparison could be further clarified.

Overall, the claim appears plausible; however, it could benefit from a more comprehensive comparison and further investigation.

**Essential References Not Discussed:**

None that I am aware of.

**Experimental Designs Or Analyses:**

I have raised my concerns in [C3] regarding the experiment design.

**Methods And Evaluation Criteria:**

I have raised some concerns regarding the baselines in [C2; C3]. Apart from them, the evaluation metrics and data seem reasonable.

**Other Comments Or Suggestions:**

None

**Other Strengths And Weaknesses:**

Strengths:

1. The novel method which proposes in-context exploration entirely without rewards
2. Well-written with accessible explanations
3. Evaluation in the real-world setting

Weaknesses:

1. No discussion or conclusion sections are found.

 Other weaknesses are listed in the section “Claims And Evidence”.

**Questions For Authors:**

I have numbered my concerns as [Concern N] in the 'Claims And Evidence' section. Discussing these concerns could potentially change my opinion and affect the score.

[Edit] I have updated my score to 4.

**Relation To Broader Scientific Literature:**

The paper introduces a novel method that learns to explore from offline data and can adapt to new tasks by using the exploration strategy. To my mind, this is the first method that do not use the rewards or learn its approximation during inference, working entirely with states and actions. However, novelty is not the only criterion for a paper to be of high quality; this paper could be improved by clarifying its claims, adding missing baselines, and justifying some of its evaluation choices.

**Theoretical Claims:**

I read the informal proposition in the main text. I did not check correctness of the proof in the Appendix A.

---

> ### Author Rebuttal · Authors · 2025-04-01
>
> We thank the reviewer for their feedback. We have added an additional unsupervised RL baseline to the [Libero experiment](https://drive.google.com/file/d/19kFpxrczDTiXL1S0rL1FfX2zXwAT1qVZ/view?usp=share_link), which we found performed worse than both BE and BC, and have also run additional experiments with OpenVLA, testing out the reviewer’s suggestion for enabling OpenVLA to explore. See below for discussion on these points and additional clarifications.
>
> ## [Concern 2] New RL Baseline on Libero
> We have added SUPE, the most effective unsupervised RL baseline, on Libero. Please see Figure 1 [here](https://drive.google.com/file/d/19kFpxrczDTiXL1S0rL1FfX2zXwAT1qVZ/view?usp=share_link). As illustrated, SUPE performs significantly worse than both BC and our approach. We note that the number of episodes we consider for Libero (15) is significantly less than what is typically required by RL algorithms to effectively learn, so we do not find this result surprising—it further illustrates the necessity for fast in-context adaptation as compared to standard RL gradient-based updates.
>
>
> ## [Concern 4] New OpenVLA Experiments
> We make several comments on our inclusion of OpenVLA as a baseline. First, our goal in including it is to demonstrate that existing state-of-the-art methods for robotic control do not effectively explore over the space of possible tasks when the goal specification is ambiguous. In other words, OpenVLA does not already solve the problem we are attempting to solve—if the specified goal is ambiguous, OpenVLA does not attempt different behaviors until it finds the correct one, it simply chooses the behavior it deems most likely.
>
> Second, as the reviewer has highlighted, OpenVLA receives no feedback on whether it has succeeded. This is fundamental to OpenVLA, however, as well as most other current state-of-the-art BC methods for robotic control: OpenVLA is not trained to condition on a history of past interactions or observations, and therefore fundamentally cannot adapt online. Our approach aims to fill precisely this gap, and we hope it will be incorporated in future policy training to enable effective adaptation to history. We note as well that, as discussed in our reply to Reviewer kepy and illustrated by our new BC with history conditioning baseline on Libero, naively conditioning on history can actually hurt performance—our approach may be a way to instead improve performance when conditioning on history.
>
> Given this, instructing OpenVLA to pick up “another” object is an ambiguous command—since OpenVLA has no history dependence, it has no grounding for what “another” object is. Nonetheless, we ran an additional trial with OpenVLA on the command “pick up the other silver pot”. We evaluate on the task illustrated in Figure 18, and give the results below, indicating which fraction of episodes the policy goes to the pot on the left, the right, or fails to grasp at either, and compare to our original command.
>
> | Command | Left | Right | Neither |
> |-|-|-|-|
> | “pick up the silver pot” | 1/8 | 7/8 | 0/8 |
> | “pick up the other silver pot” | 2/8 | 4/8 | 2/8 |
>
> As this illustrates, while this command does cause OpenVLA to move to the left pot somewhat more frequently, it also hurts its ability to move to either pot at all (likely because this command is somewhat outside the training distribution for OpenVLA).
>
> ## [Concern 1] Ease of Training and Evidence for Faster Adaptation
> To clarify our claim that the method is easier to train, this is indeed somewhat tangential to the claim of faster exploration and adaptation online, the primary claim of our paper and the main focus of our experiments. However, our approach does lead to lower computational overhead and simplicity at deployment compared to the baseline RL methods, all of which require online gradient updates and hyperparameter tuning. While this is an advantage of our approach, we see this as complementary to our main objective.
>
> We would also like to emphasize that BE is able to explore significantly faster on Antmaze and Libero than all RL baselines, supporting our main claim of faster adaptation and exploration. For example, to reach the final performance of SUPE, the best RL baseline, it requires only 15% of the samples. While the performance between BE and the baseline RL methods is similar on Franka Kitchen, we believe the strong performance on Antmaze and Libero highlights that in many settings, our approach leads to a substantial gain.
>
> ## [Concern 3] Scene Conditioning
> As noted in the paper, Libero 90 consists of 90 tasks distributed across 21 scenes. The scene conditioning vector is then simply a one-hot vector of dimension 21 indicating which scene the agent is operating in. While this information can be inferred entirely from visual information (the scenes appear visually distinct) we found that adding this conditioning improved the performance of both our approach and BC slightly. This is not critical to performance, however.

---

### Official Review · Reviewer_kepy · 2025-03-14

**Overall Recommendation:** 2

**Summary:**

This paper addresses the problem of in-context RL -- in particular, learning how to *explore* through in-context adaptation.

The proposed method (Behavioral Exploration, BE) performs reweighted behavior-cloning over large expert datasets, with a long-context policy that also conditions on a variable representing how exploratory the expert’s behavior is.

The pretrained BE policy is validated in downstream RL / IL tasks, and demonstrates improved performance over standard BC and RL approaches.

**Claims And Evidence:**

A key claim is that BE learns to explore over the space of expert behaviors from an expert demonstration dataset, which should speed learning on downstream tasks.

- The experiments, however, do not evaluate BE as a pretraining method for any downstream tasks.


Another key claim is that BE performs in-context adaptation.

- The provided explanations and experiments do not totally convince me that BE is adaptively exploring in-context.

    - The reuslts on D4RL show that BE learns a better policy and explores more than baselines, but do not demonstrate any *adaptive* exploration ability -- these results could also occur simply because BE is better at modeling the behaviors shown in the offline dataset.

    - I would be more convinced if the authors demonstrate that BE is able to learn faster on a completely unseen task than baselines.

**Essential References Not Discussed:**

None that I am aware of.

**Experimental Designs Or Analyses:**

yes. See claims.

**Methods And Evaluation Criteria:**

The authors consider naive BC as a baseline, conditioning on state only. How well would BC with the diffusion transformer policy backbone + conditioning on history, but without the reweighting objective, perform?

**Other Comments Or Suggestions:**

Typo in Line 880: affective -> effective

**Other Strengths And Weaknesses:**

Strengths

- Overall, the paper seems well motivated

- The proposed approach is based on BC, making it applicable for robotics settings.

- Strong empirical performance in both simulated and real robot experiments


Weaknesses

- There are some places where a lot of jargon is used, making it difficult to grasp what the authors mean:

    - In particular, the section on selecting the history distribution (pg 5) is quite confusing.

    - L257: what is the induced distribution of online states? Induced by what?

    - Why is estimating the distribution of online states a fixed point problem? How is the fixed point problem speciied?

- In-context learning is not clearly defined. What criteria should be used to determine if a policy has successfully adapted its behavior online?

**Questions For Authors:**

1. The notation in the paper suggests that the demonstration data is generated by a single expert, but the text argues that the demonstration dataset should contain multiple behaviors. Is there any reason why the demo dataset can only contain behavior from one expert?

2. How is the coverage objective, which attempts to cover behaviors shown in $pi_\beta$ diffrent from imitation learning approaches such as feature matching?

3. How can we decouple the performance improvement due to learning to explore, versus the performance improvement due to access to demonstration data?

**Relation To Broader Scientific Literature:**

This paper examines how to equip a policy with exploration abilities based on contextual information. This relates to the emerging area of in-context RL and meta-RL.

**Theoretical Claims:**

I did not check the theory.

---

> ### Author Rebuttal · Authors · 2025-04-01
>
> We thank the reviewer for their feedback. We have run an additional requested baseline on Libero—BC with history conditioning, which we found performed worse than BC—and additional experiments illustrating that BE adapts to its history online; please see below for further discussion and the results [here](https://drive.google.com/file/d/19kFpxrczDTiXL1S0rL1FfX2zXwAT1qVZ/view?usp=share_link).
>
> ## New Experiment Showcasing BE’s Ability to Adapt Online
> Please see this Figures 2 and 3 [here](https://drive.google.com/file/d/19kFpxrczDTiXL1S0rL1FfX2zXwAT1qVZ/view?usp=share_link) for illustration of the BE’s ability to adapt to history. To illustrate adaptivity, we run a BE policy trained on Antmaze Large, conditioning on either (a) history sampled from states visited in past online episodes (the strategy used in our experiments), or (b) histories containing only a single fixed trajectory. As our results illustrate, the history can have significant impact on the agent’s performance: when conditioning on a single trajectory, the agent largely takes routes that avoid the space traversed by this trajectory, adapting its behavior to visit novel states. Given this, we see that the agent achieves significantly higher coverage conditioning on the past states it has observed, rather than a single fixed trajectory.
>
> ## New Experiment Showing History Conditioning Hurts Standard BC
> We have a trained a BC policy with history conditioning on Libero—please see Figure 1[here](https://drive.google.com/file/d/19kFpxrczDTiXL1S0rL1FfX2zXwAT1qVZ/view?usp=share_link). As can be seen, conditioning the BC policy on history actually hurts performance. We believe this is an example of “causal confusion” [1], a known phenomenon where conditioning on additional information can decrease performance of BC by introducing spurious correlations.
>
> ## Learning on Downstream Tasks
> The reviewer states that: “A key claim is that BE learns... The experiments... downstream tasks”. We want to highlight that the main focus of the paper is not training on downstream tasks, but rather pretraining a policy that can explore effectively online, focusing its exploration on behaviors exhibited by the demonstration data. While existing work (e.g. [2]) has shown that a richer supply of data does improve policy learning on downstream tasks—and given this we believe that data collected with BE may indeed be useful for learning downstream tasks—this is not the main focus of the paper. We are instead primarily interested in how to pretrain a policy for exploration, and our experiments were chosen to reflect this. We will update the exposition in the paper to make this clear.
>
> ## Additional Clarifications
> *History Distribution*: At deployment, we condition the BE policy on the history of states it has already collected online. As any policy $\pi$ induces some distribution over states visited, the distribution of history encountered at test time depends on the learned BE policy. Ideally, we would like the distribution the BE policy was trained on to match this online distribution. As described in Section 4.3, this is challenging as we do not know what the state visitation distribution of the learned policy will be. Addressing this is a fixed-point problem in that we must first optimize a BE policy on some history distribution, deploy it to find what history distribution it induces, then re-optimize a policy on this new history distribution, and repeat until convergence. To avoid this complication, we suggest choosing the history distribution in training to be a uniform distribution over trajectories in the offline dataset, which we found worked well in practice.
>
> *Number of Demonstrators*: The offline data may be collected by different demonstrators; in this case $\pi_{\beta}$ is a mixture policy over all demonstrator policies.
>
> *How can we decouple the performance improvement…?*:
> While our approach can learn to explore effectively with any sufficiently diverse set of offline data, it focuses its exploration on the behaviors exhibited in the offline data. Therefore, while diverse low-quality data may be sufficient for training a policy that collects diverse data, it is not necessarily sufficient for successfully achieving a task; the demonstration data must contain examples of successful task completion for BE to learn these behaviors. Thus, both the learning to explore and the type of demonstration data are important for our approach.
>
> *How is the coverage objective…*: Would the reviewer be able to provide additional clarification for what they mean by “feature matching” here? One potential difference is that BE aims to induce *different* behaviors, while typical BC usually aims to induce the *same* behaviors.
>
> [1] De Haan, Pim, et al. "Causal confusion in imitation learning." NeurIPS, 2019.
>
> [2] Yarats, Denis, et al. "Don't change the algorithm, change the data: Exploratory data for offline reinforcement learning." arXiv preprint arXiv:2201.13425 (2022).

---

### Official Review · Reviewer_fhX3 · 2025-03-17

**Overall Recommendation:** 3

**Summary:**

The manuscript presents "Behavioral Exploration" (BE), a novel formalization for learning from expert demonstrations, while giving the ability to the policy/user to explore in a controlled manner (via some parameters). The main innovation is the Behavior Policy Coverage metric, and how to use it to condition on previous historical data to enhance exploration (or exploitation). The authors provide a practical algorithm and the results showcase that BE outperforms several baselines in RL and imitation learning scenarios.

**Claims And Evidence:**

Overall the authors' claims are matched and supported by the evidence provided. The authors provide proof for the main theoretical result, and many experiments and explanations for the empirical results.

**Essential References Not Discussed:**

I did not find any.

**Experimental Designs Or Analyses:**

I have a few comments here (tbh I was confused if those comments fit this section or the section on "Methods And Evaluation Criteria"):

1) The RL setting is not described at all (the explanation in the supplementary is not enough). I understand that there is a lack of space, but we need to be more specific here. It is implied that the authors do something similar with two other papers, but we never see the exact pipeline. This is important as the authors might not even do what the other papers describe. Reading the paper alone (and not the other papers), it is not clear how the RL scenarios are set up. In other words, the policy learned by BE is used only for exploration? Is there policy adaptation based on newly acquired rewards/data? How is this incorporated with the BE policy? Overall, this part needs more explanation.

2) I am not sure I understand the real-world experiment. The authors say that this is "a reaching task", but then they say "we count a trial as a success if the policy interacts with both objects". Again the setting here is poorly described.

3) The task success definitions and metrics (e.g. "we count a trial as a success if the policy interacts with both objects") are quite vague. Thus, this makes us skeptical about the results. If they are actually not so vague (but due to lack of space, the authors had to come up with a sentence), then they need to be defined more precisely in the text (or even supplementary). Otherwise, the authors should perform new experiments with more rigorous metrics.

4) Aggregated performance curves can lead to misleading results. The authors should at least describe how they aggregated the results from different tasks.

**Methods And Evaluation Criteria:**

The D4RL and LIBERO datasets/scenarios and the real world experiments are suited to evaluate the task and provide meaningful comparisons overall. The RL and real-world experiments require some clarifications (see "Experimental Designs Or Analyses"), but otherwise the experiments are well-suited for validating the algorithm.

**Other Comments Or Suggestions:**

Very well written paper.

**Other Strengths And Weaknesses:**

Strengths
-------------

- Well written paper
- The objective is clearly conveyed
- The BE policy learning pipeline is nice and imho novel

Weaknesses
-----------------

- The authors could "give up" a few experiments (e.g. D4RL kitchen) and provide more meaningful discussions of the results
- The aggregated performance curves

**Questions For Authors:**

- How would BE work in tasks where we require more fine control? E.g. when we require torque control? Or low-level joint space control? (The ant maze experiment does not require fine control imho)
- What would be some more quantitative metrics for evaluating performance? "we count a trial as a success if the policy interacts with both objects" (and other similar metrics in the paper) are quite vague and cannot really be trusted

**Relation To Broader Scientific Literature:**

The ability to effectively explore vast spaces can beneficial in many fields. Moreover, making it possible to "ground" exploration to real-world experience (via demonstrations and the conditioning) can be even more useful and beneficial, since it can shrink the exploration time while not losing interest parts of the state space. Overall, the problem tackled by the paper is very interesting, and the paper provides one solution to this.

**Theoretical Claims:**

I did not check the correctness thoroughly, but the claim and proof make sense.

---

> ### Author Rebuttal · Authors · 2025-04-01
>
> We thank the reviewer for their comments. In the following we provide additional clarification on the success metrics and RL experiments, as well as several other comments the reviewer had. We will update the final version of the paper to include all these details.
>
> ## Clarification of Success Metrics
> For the real-world WidowX experiments, we define an “interaction” with an object as the end-effector of the WidowX making contact with the object (as stated in the supplemental: “We consider a reach to be successful if the end-effector makes contact with the object”). We note that in the settings we considered this is straightforward and unambiguous to evaluate. As we are evaluating the ability of each approach to explore the possible tasks in the scene, we run the policy 5 times, and count the overall sequence as a success if across these 5 trials both objects in the scene were interacted with—the end-effector made contact with each object at least once across the 5 episodes.
>
> For Libero and D4RL Kitchen, we utilize the built-in success detector in the simulator to identify success. For D4RL Antmaze, we utilize the same goal locations as in (Wilcoxson et al., 2024), and count a goal as reached if the agent reaches within distance 0.5 of a square of side length 1.5 centered at the goal point. For each setting, we evaluate on each task separately (e.g. each goal for Antmaze, or task for Libero), and the plots given in the paper are performance averaged across each task (see below for further clarification on aggregation of results). Figures 2, 4, 6, 7 therefore show what fraction of trials have resulted in a success up to a particular time (averaged across tasks and random seeds).
>
> We will add all these details to the final version of the paper. If the reviewer has any additional questions on the success metrics we are happy to provide further clarification.
>
> ## Clarification on RL Problem Setting and Experimental Details
> For all RL baselines, we run them as described in their original works. In particular, for each setting the algorithm is given an offline dataset of transitions from the environment, but with no reward labels. With the exception of “Online RND”, each method we consider pretrains on this data (Online RND only trains online). After pretraining, each method is deployed online, and is evaluated on the number of steps it takes to reach the specified goal at least once.
>
> The BE policy is therefore used only for exploration, and is not fine-tuned at all during online deployment. However, as described in Section 4, the history of previous (online) interactions is fed into the context of the BE policy online, so that its behavior can adapt to the past experience—this adaption is only “in-context”, however; no gradient steps are taken on the BE policy. For every other approach considered, with the exception of BC (which is not updated at all online), the policy is updated online with standard gradient-based RL updates (in particular, which seek to maximize RND bonuses, inducing exploration).
>
> ## Aggregated Performance Curves
> We observe similar trends in the individual results as in the aggregated results, but for the final version will include in the upplemental individualized results on Antmaze (for each maze type and goal location) and Libero (for each of the 90 tasks). For the WidowX results, the supplemental already provides individualized results (see Figures 20-22). In all cases where results were aggregated (Antmaze, Libero, and WidowX settings), the aggregated results are the mean of the individual results.
>
> ## Fine-Grained Control
> In addition to Antmaze, which, as noted by the reviewer, does require joint-space control, the Franka Kitchen environment considered in Figure 4 also requires joint-space control. We believe that these results indicate that our method does scale effectively to settings with joint-space control.
>
> More generally, we would expect our method to scale effectively to any setting on which BC performs well. As our method is essentially training on a BC objective augmented with a particular conditioning structure, we would expect it to, in general, perform at least as well as BC, which we found to be the case in all our experiments. The degree to which it learns effective exploratory behavior is a function of several additional factors, in particular the distribution of behaviors present in the offline data, so there may be cases where it is not able to learn more effective exploratory behavior than BC—we do not necessarily expect the type of control to affect this, however.

---

### Decision · Program_Chairs · 2025-05-01

**Decision:**

Accept (poster)

**Comment:**

The paper introduces "Behavioral Exploration", where an offline demonstration dataset is used to train a policy that can condition on how "exploratory" it is. The connection to in-context learning is that the policy conditions on a long history. Results on standard benchmarks as well as an actual physical robot provide empirical support for the method.

Although not all the reviewers support publication at this time, upon closer inspection, it's not clear that the specific criticisms should doom the paper: "a lot of jargon" [kepy] and one who cannot "agree with the motivations" [RYNM]. The other positive reviewers (and myself) find value in the motivation of the work, and it is normal for researchers to sometimes disagree about the significance over particular problem statement/motivation.

Therefore, I recommend accepting the paper.